# A Statistical Learning Perspective on Semi-dual Adversarial Neural Optimal Transport Solvers

**Roman Tarasov**[1], **Petr Mokrov**[1], **Milena Gazdieva**[1,2], **Evgeny Burnaev**[1,2], **Alexander Korotin**[1,2]
[1] Applied AI Institute, *Moscow, Russia*
[2] AXXX, *Moscow, Russia*
`tarasov.rb@yandex.ru`

## Abstract

Neural network-based optimal transport (OT) is a recent and fruitful direction in the generative modeling community. It finds its applications in various fields such as domain translation, image super-resolution, computational biology and others. Among the existing OT approaches, of considerable interest are adversarial minimax solvers based on semi-dual formulations of OT problems. While promising, these methods lack theoretical investigation from a statistical learning perspective. Our work fills this gap by establishing upper bounds on the generalization error of an approximate OT map recovered by the minimax quadratic OT solver. Importantly, the bounds we derive depend solely on some standard statistical and mathematical properties of the considered functional classes (neural nets). While our analysis focuses on the quadratic OT, we believe that similar bounds could be derived for general OT case, paving the promising direction for future research. Our experimental illustrations are available online https://github.com/milenagazdieva/StatOT.

## 1 Introduction

In recent years, there has been a boom in the development of computational Optimal Transport (OT) which has been identified as a powerful tool of solving various machine learning problems, e.g., biological data transfer (Bunne et al., 2023; Koshizuka & Sato, 2022; Vargas et al., 2022), image generation (Wang et al., 2021; De Bortoli et al., 2021; Chen et al., 2021) and domain translation (Xie et al., 2019; Fan et al., 2023) tasks. The first works in the OT field had proposed methods for solving OT problems between discrete distributions (Cuturi, 2013; Peyré et al., 2019). The next milestone was the emergence of OT-based neural methods using the OT cost as a loss function for updating the generator in generative models (Sanjabi et al., 2018; Gulrajani et al., 2017; Petzka et al., 2018; Liu et al., 2019). Only recently, the ML community has experienced an explosion of interest in developing scalable neural methods which compute OT map (or plan) and use it directly as the generative model, see, e.g., (Rout et al., 2022; Daniels et al., 2021). Methods of this kind are usually attributed as *continuous OT solvers* and typically have minimax optimization objectives based on semi-dual formulations of OT problems.

Although a large number of neural network based *minimax* OT solvers have already been developed, e.g., (Fan et al., 2023; Korotin et al., 2023b;a; Nguyen et al., 2024; Buzun et al., 2025; Gazdieva et al., 2023), there is a relatively limited understanding of the theoretical aspects of their losses. Existing theoretical investigations (Makkuva et al., 2020; Rout et al., 2022; Fan et al., 2023; Asadulaev et al., 2024; Kolesov et al., 2024a) do not reveal practical convergence guarantees, i.e., statistical rates, of these solvers. In contrast, several works on *non-minimax* OT solvers (Gunsilius, 2022; Hütter & Rigollet, 2021) develop statistical bounds for their solutions. However, the analysis of minimax solvers is significantly more complex as their optimization objective includes two variables, OT map and separate dual variable, while non-minimax approaches just operate with only one variable.

**Contributions.** In our paper, we fill the aforementioned gap by conducting a thorough theoretical investigation of minimax quadratic OT solvers. *First*, we show that for the minimax OT solvers, the *generalization error*, i.e., the difference between true and approximate OT maps, can be upper

bounded by the sum of estimation and approximation errors (Theorem 4.1). *Second*, we specify the bounds for the estimation errors by showing that they depend only on the Rademacher complexities of the classes of the neural nets used (Theorem 4.2). This result highlights the importance of *consistent estimation* theory in the context of minimax OT solvers. *Third*, we reveal the *approximation ability* of OT solvers, i.e., show that the approximation errors can be made arbitrarily small by choosing the appropriate classes of neural nets (Theorems 4.3, 4.6). As a result, we establish the *learnability guarantees* for minimax OT solvers by showing that their generalization errors also can be made arbitrarily small by choosing the sufficient number of samples and appropriate classes of neural nets (Theorem 4.9, Corollary 4.10). We support our theoretical results with an illustrative experiment, §5.

**Notations.** In the paper, $\mathcal{X}, \mathcal{Y} \subset \mathbb{R}^D$ are convex compact subsets; continuous scalar-valued functions on $\mathcal{X}$ are $\mathcal{C}(\mathcal{X})$; absolutely continuous prob. distributions on $\mathcal{X}$ are $\mathcal{P}_{ac}(\mathcal{X})$. We use the same symbol "$p$" to denote a continuous distribution $p \in \mathcal{P}_{ac}(\mathcal{X})$ and its density $p(x)$ at some point $x \in \mathcal{X}$. For a measurable map $T : \mathcal{X} \to \mathcal{Y}$, we use $T_{\#}$ to denote the associated push-forward operator. Throughout the text, $\| \cdot \|_{L^2(p)}$ denotes the $L^2$ norm w.r.t probability measure $p$, i.e., for $T_1, T_2 : \mathcal{X} \to \mathcal{Y}$:

$$\|T_1 - T_2\|_{L^2(p)}^2 = \int_{\mathcal{X}} \|T_1(x) - T_2(x)\|_2^2 p(x) dx.$$

For a function $\varphi \in \mathcal{C}(\mathcal{Y})$, its convex conjugate is $\overline{\varphi} : \mathcal{X} \to \mathbb{R}$; $\overline{\varphi}(x) \stackrel{\text{def}}{=} \max_{y \in \mathcal{Y}} \{\langle x, y \rangle - \varphi(y)\}$ (similarly for functions supported on $\mathcal{X}$). Note that convex conjugate is always a convex function.

## 2 BACKGROUND AND OBJECT OF STUDY

In this section, we provide key concepts of OT theory that are used in our paper, a comprehensive introduction could be found in (Santambrogio, 2015; Villani, 2008). Throughout the paper, $p \in \mathcal{P}_{ac}(\mathcal{X})$ and $q \in \mathcal{P}_{ac}(\mathcal{Y})$ are absolute continuous source and target distributions.

**Monge's OT problem.** Given $p, q$ and a continuous cost function $c : \mathcal{X} \times \mathcal{Y} \to \mathbb{R}$, Monge's primal OT prescribes to find a measurable transport map $T : \mathcal{X} \to \mathcal{Y}$ which minimizes OT cost:

$$\text{Cost}_c(p, q) \stackrel{\text{def}}{=} \inf_{T_{\#}p=q} \int_{\mathcal{X}} c(x, T(x)) p(x) dx. \tag{1}$$

In other words, we want to find a map $T$ that transports probability mass from $p$ to $q$ in the cheapest way with respect to the given cost function $c(x, y)$, see Fig. 1. I.e., the average transport expenses when moving $p$ to $q$ should be minimal.

**Wasserstein-2 distance.** A popular example of an OT cost is the (squared) Wasserstein-2 distance ($\mathbb{W}_2^2$), for which the cost function is quadratic, i.e., $c(x, y) = \frac{1}{2}\|x - y\|_2^2$. In this case, the corresponding OT objective (1) is known to permit the unique minimizer $T^*$.

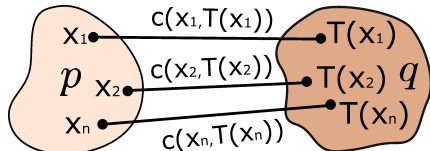

Figure 1: Monge's formulation of optimal transport.

**Semi-dual OT problem for quadratic cost**. Primal OT problem (1) with quadratic cost $c(x, y) = \frac{1}{2}\|x - y\|_2^2$ has a tight dual counterpart[1] (Santambrogio, 2015, Thm. 1.39):

$$\text{Cost}_{\frac{1}{2}\|\cdot - \cdot\|_2^2}(p, q) = \int_{\mathcal{X}} \frac{\|x\|_2^2}{2} p(x) dx + \int_{\mathcal{Y}} \frac{\|y\|_2^2}{2} q(y) dy - \min_{\varphi \in \mathcal{C}(\mathcal{Y})} \left\{ \int_{\mathcal{X}} \overline{\varphi}(x) p(x) dx + \int_{\mathcal{Y}} \varphi(y) q(y) dy \right\}, \tag{2}$$

where $\overline{\varphi}$ is the convex conjugate of a potential $\varphi \in \mathcal{C}(\mathcal{Y})$. Potential $\varphi^*$ which delivers maximum to (2) is called *Kantorovich* potential. Removing in (2) first two terms that do not depend on $\varphi$, we get an equivalent formulation:

$$\mathcal{L}(\varphi) \stackrel{\text{def}}{=} \min_{\varphi \in \mathcal{C}(\mathcal{Y})} \int_{\mathcal{X}} \overline{\varphi}(x) p(x) dx + \int_{\mathcal{Y}} \varphi(y) q(y) dy. \tag{3}$$

The optimal potential $\varphi^*$ which solves (3) is known to be convex ((Santambrogio, 2015, Proposition 1.21)). Therefore, we can optimize (3) with respect to the set of the continuous convex potentials $\varphi \in \text{Cvx}(\mathcal{Y}) \subset \mathcal{C}(\mathcal{Y})$:

---

[1]Strictly speaking, (2) appears as the dual for *Kantorovich's* relaxation of (1), (Santambrogio, 2015, §1.1). However, (1) = (2) still holds (for the quadratic cost, $c(x, y) = \frac{1}{2}\|x - y\|_2^2$), see (Santambrogio, 2015, Thm. 1.33)

$$\mathcal{L}(\varphi) = \min_{\varphi \in \text{Cvx}(\mathcal{Y})} \int_{\mathcal{X}} \overline{\varphi}(x) p(x) dx + \int_{\mathcal{Y}} \varphi(y) q(y) dy. \tag{4}$$

Furthermore, we can recover the OT map $T^*$ which solves (1) for quadratic (scalar product) cost from optimal dual potential $T^*(x) = \nabla \overline{\varphi^*}(x)$ (Santambrogio, 2015, Thm. 1.17).

**Continuous OT problem.** Analytical solutions for problems like (2) are, in general, not known. In real-world scenarios, the measures $p, q$ are typically not available explicitly but only through their empirical samples $X = \{x_1, \ldots, x_N\} \sim p$, $Y = \{y_1, \ldots, y_M\} \sim q$. To approximate the desired solution for primal (dual) OT, two setups are possible. In the first setup called **discrete** (Peyré et al., 2019), one aims to establish optimal matching (point-to-point correspondence) between the empirical distributions $\widehat{p} = \frac{1}{N} \sum_{n=1}^{N} \delta_{x_n}$, $\widehat{q} = \frac{1}{M} \sum_{m=1}^{M} \delta_{y_m}$. In turn, our study focuses on the second **continuous** setup. Under this setup, the aim is to recover some parametric approximation $\widehat{T}$ of OT map $T^*$ based on available training samples from the reference distributions $p, q$. Compared to discrete OT, continuous OT allows easy estimation of the recovered map $\widehat{T}(x^{\text{new}})$ for new (unseen) samples $x^{\text{new}} \sim p$, see Fig. 2. In contrast, discrete OT solvers perform matching between the empirical distributions and, *by default*, do not provide *out-of-sample* estimation of the transport map. Naive interpolation techniques (neural network regression on top of the discrete OT solutions) are known to provide unsatisfactory results, see (Gazdieva et al., 2023, Appendix D), for the illustration of this issue. While several recent papers (Klein et al., 2024; Deb et al., 2021; Manole et al., 2024; Pooladian & Niles-Weed, 2021) propose more advanced interpolation techniques and develop out-of-sample estimators, their applicability in high dimensions requires a separate study.

**Continuous OT solvers for quadratic cost.** In recent years, numerous algorithms have been developed to solve the continuous optimal transport problem for the quadratic cost. The primal problem is challenging to solve due to the difficulty in satisfying the constraint $T_\# p = q$. For this reason, semi-dual problem (3) or (4) is solved instead and optimal transport map

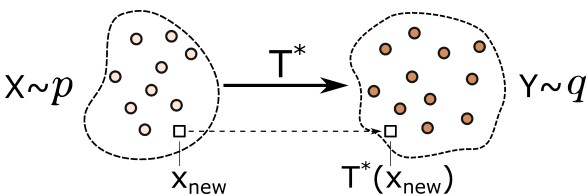

Figure 2: Continuous setup of OT problem.

$T$ is recovered from the optimal dual potential. One popular strategy (Taghvaei & Jalali, 2019) is to consider (4) and parameterize learned $\varphi$ as *input convex neural network* (ICNN) (Amos et al., 2017). ICNNs are neural networks which place certain restrictions on their weights and activations to ensure the convexity w.r.t. the input. Given proper parametrization of dual potential $\varphi_\theta, \theta \in \Theta$, one can directly minimize:

$$\mathcal{L}(\varphi_\theta) = \int_{\mathcal{X}} \overline{\varphi_\theta}(x) p(x) dx + \int_{\mathcal{Y}} \varphi_\theta(y) q(y) dy.$$

or its minibatch (Monte-Carlo estimate) version. Such methods, however, require additional optimization steps to compute $\overline{\varphi_\theta}$, which may be undesirable from the computational efficiency perspectives. To overcome the issue, researchers have explored ways to simplify the task. Several studies propose methods that approximate the conjugate potential using another neural network (Makkuva et al., 2020; Korotin et al., 2021a; Amos, 2023). Such methods, however, may suffer from *gradient deviation* issue (Korotin et al., 2021b, §2) and fail to recover OT mappings sufficiently well.

**Continuous minimax OT solvers for quadratic cost.** Recently, minimax semi-dual approaches have been actively explored, they learn both the dual potential and the primal transport map. This formulation can be derived by applying the interchange theorem (Rockafellar, 2006, Theorem 3A):

$$\int_{\mathcal{X}} \overline{\varphi}(x) p(x) dx = \int_{\mathcal{X}} \max_{y \in \mathcal{Y}} \{\langle x, y \rangle - \varphi(y)\} p(x) dx = \max_T \int_{\mathcal{X}} [\langle x, T(x) \rangle - \varphi(T(x))] p(x) dx, \tag{5}$$

the outer $\max_T$ is taken w.r.t. measurable maps $T : \mathcal{X} \to \mathcal{Y}$.

*Remark.* The existence of a map which maximizes the above problem follows from the measurable selection theory. For $\varphi \in \mathcal{C}(\mathcal{Y})$, the sets $\arg\max_y \{\langle x, y \rangle - \varphi(y)\}$, $x \in \mathcal{X}$ are nonempty and closed. From (Aubin & Frankowska, 2009, Thm. 8.1.3), it follows that there exists a measurable map $T$ with values in such sets. This map delivers maximum to (5).

In light of (5), problem (3) is thus reformulated as a minimax optimization problem:

$$(3) = \min_{\varphi} \max_{T} \left\{ \int_{\mathcal{X}} [\langle x, T(x) \rangle - \varphi(T(x))] p(x) dx + \int_{\mathcal{Y}} \varphi(y) q(y) dy \right\} \stackrel{\text{def}}{=} \min_{\varphi} \max_{T} \mathcal{L}(\varphi, T). \quad (6)$$

Under certain assumptions one may guarantee that if the values $\mathcal{L}(\varphi, T)$ and $\mathcal{L}(\varphi^*, T^*)$ are close, then $T$ is close to OT map $T^*$. And it is our work which establishes **practical convergence guarantees** for (6) under continuous OT setup with $\varphi, T$ given by neural networks, see §4.

Importantly, the use of minimax semi-dual OT losses like (6) is an emerging direction in generative modeling (Korotin et al., 2021b; Rout et al., 2022; Gazdieva et al., 2025; Korotin et al., 2023a;b; Fan et al., 2023; Gazdieva et al., 2023; Choi et al., 2024a;b; Buzun et al., 2025; Asadulaev et al., 2024; Nguyen et al., 2024; Carrasco et al., 2023; Kolesov et al., 2024a), making the analysis of (6) highly beneficial and desirable to the community. In particular, some of the minimax OT works, e.g., (Korotin et al., 2021b; Rout et al., 2022), explore quadratic (scalar product) cost functions, i.e., deal exactly with (6). The others consider more general cost functions (Fan et al., 2023) and OT formulations (Asadulaev et al., 2024). Extending the results of our study for such setups is a fruitful direction of future research.

**Object of study.** In practice, $T$ and $\varphi$ are parametrized as neural networks $T_\omega$, $\omega \in \Omega$ and $\varphi_\theta$, $\theta \in \Theta$. Besides, following our continuous OT setup, the (unknown) reference distributions $p, q$ are replaced by their empirical counterparts $\widehat{p}, \widehat{q}$. This yields the optimization problem typically solved in practice:

$$\min_{\theta \in \Theta} \max_{\omega \in \Omega} \sum_{n=1}^{N} \frac{\langle x_n, T_\omega(x_n) \rangle - \varphi_\theta(T_\omega(x_n))}{N} + \sum_{m=1}^{M} \frac{\varphi_\theta(y_m)}{M} \stackrel{\text{def}}{=} \min_{\theta \in \Theta} \max_{\omega \in \Omega} \widehat{\mathcal{L}}(\varphi_\theta, T_\omega). \quad (7)$$

ML practitioners and researchers naturally wonder how different are the practical map $\widehat{T}_{\omega^*}$ that solves (7) and OT map $T^*$. Formally speaking, the problem is to estimate the *generalization error*

$$\mathbb{E}_{X,Y} \left\| T^* - \widehat{T}_{\omega^*} \right\|^2_{L^2(p)}, \quad (8)$$

where the expectation is taken w.r.t. empirical training samples $X \sim p, Y \sim q$. Remarkably, the presence of the error (8) comes from two sources: a) reliance on the restricted families of optimized potentials, maps (*approximation error*); b) usage of empirical measures $\widehat{p}, \widehat{q}$ instead of original ones (*estimation error*). Establishing the particular bounds for (8) is our primal focus, see §4.

## 3 RELATED WORKS

In this section, we discuss the existing works which conduct theoretical analysis of semi-dual OT losses. The current progress in this field could be subdivided into two branches reviewed below.

**Non-minimax semi-dual OT.** The first branch of works (Hütter & Rigollet, 2021; Gunsilius, 2022) analyze the non-minimax losses (3) and (4) and develop error bounds (8) for pushforward maps $T$ given by the gradient of dual potentials, i.e., $\mathbb{E}_{X,Y} \|\nabla \overline{\varphi}^* - \nabla \widehat{\overline{\varphi}}_\omega\|^2_{L^2(p)}$. To achieve particular statistical rates, the authors place certain restrictions on the considered problem setup and deal with specific classes of functions (maps), e.g., wavelet expansions or (kernel-smoothed) plug-in estimators. Recent studies in this direction (Divol et al., 2025; Ding et al., 2024) extend the analysis of error rates for the class of potentials $\varphi$ given by neural networks, e.g., input convex neural networks. Meanwhile, none of the mentioned works treats the learned map $T$ separately from dual potential $\varphi$. Importantly, the analysis of minimax objective (6) is considerably more challenging than (3) due to the additional "degree of freedom" given by the optimized map $T$. Furthermore, saddle-point problems such as (6) are known to be more tricky for theoretical investigation than usual minimization. At the same time, the practical demand stemming from the recent proliferation of minimax-based OT solvers, see §2, makes such an investigation highly desirable. All of this necessitates separate study of minimax OT solvers; the adaptation of existing non-minimax results is questionable, if that is even possible.

**Minimax semi-dual OT.** The estimation of (8) in case of minimax OT (7) is a much less explored task. In fact, there are no studies at all that provide statistical bounds for recovered minimax OT map $\widehat{T}_\omega$. The existing works (Makkuva et al., 2020; Rout et al., 2022) only conduct an analysis of $\|T^* - \widehat{T}_\omega\|^2_{L_2(p)}$ in the sense of *duality gaps*. That is, the error between the true OT map and recovered

map is upper-bounded by values of functional $L(\varphi, T)$. The duality gaps analysis helps to validate minimax methodology; it is common for more broad class of (non-quadratic) minimax OT solvers (Fan et al., 2023; Asadulaev et al., 2024; Kolesov et al., 2024a). However, we again emphasize that it does not reveal particular statistical rates and guarantees for (8). In (González-Sanz et al., 2022), the authors establish promising results for the error of recovered approximate OT map. However, their problem formulation of minimax OT differs from ours, eq. (6), as they aim to treat OT as the limit of regularized GAN objectives. Finally, the work (Nietert & Goldfeld, 2025) conducts theoretical analysis of the error of solving the minimax Wasserstein distributionally robust optimization problem which again differs from ours.

**Theoretical analysis of other OT formulations.** For the completeness of exposition, we also mention several studies that develop statistical learning analysis of recovered OT map (plan) error for non semi-dual OT formulations. The works (Genevay et al., 2019; Rigollet & Stromme, 2022; González-Sanz & Hundrieser, 2023; Mokrov et al., 2024; Goldfeld et al., 2024; Korotin et al., 2024; Kolesov et al., 2024b) deal with Entropy-regularized OT (Gushchin et al., 2023); (Vacher & Vialard, 2022; 2023; Gazdieva et al., 2024) investigate unbalanced OT versions; (Benamou & Brenier, 2000) considers dynamic OT. Although these works are interesting and insightful, their object of study is different from ours, making them less relevant.

Since the minimax objective (6) resembles adversarial training, we additionally discuss the literature addressing the *theoretical aspects* of Generative Adversarial Neural Networks (GAN) losses and their *relation* to semi-dual OT losses in Appendix E. For a comprehensive analysis of the connections and differences between these losses, we refer the reader to (Gazdieva et al., 2025).

## 4 THEORETICAL RESULTS

In real-world use cases, OT practitioners are given (i.i.d.) training samples $X \sim p$ and $Y \sim q$ and optimize empirical minimax objective (7) with respect to restricted classes of functions $\varphi \in \mathcal{F}$, $T \in \mathcal{T}$, e.g., neural networks. Below we denote the solutions of the problem which we have in practice (superscript "R" stands for "Restricted"):

$$\widehat{\varphi}^R = \underset{\varphi \in \mathcal{F}}{\operatorname{argmin}} \max_{T \in \mathcal{T}} \widehat{\mathcal{L}}(\varphi, T); \quad \widehat{T}^R = \underset{T \in \mathcal{T}}{\operatorname{argmax}} \widehat{\mathcal{L}}\left(\widehat{\varphi}^R, T\right).$$

Note that in these equations we implicitly assume the *existence* of optimizers $\widehat{\varphi}^R \in \mathcal{F}, \widehat{T}^R \in \mathcal{T}$. While in general this may not always hold true, some natural practical choices of $\mathcal{F}$ and $\mathcal{T}$, e.g., neural network architectures $\varphi_\theta, T_w$ with bounded set of parameters $\Theta, \Omega$ guarantee the existence.

Our goal is to estimate the *generalization error*, i.e., the (average) error between OT map $T^*$ and empirical map $\widehat{T}^R$:

$$\underset{X,Y}{\mathbb{E}} \left\| \widehat{T}^R - T^* \right\|_{L^2(p)}. \tag{9}$$

We subdivide the problem into several steps. First, we upper-bound the error using differences in the functional $\mathcal{L}(\varphi, T)$ values (§4.1). The obtained upper bound decomposes into several terms: *estimation* and *approximation* errors that occur in both the *inner* and *outer* optimization problems within our minimax objective (6). Second, we estimate each term individually using suitable techniques from statistical learning theory (§4.2). Finally, we bring it all together and formulate our main theoretical result (§4.3). We provide the *proofs* of all theoretical results in Appendix A.

### 4.1 ERROR DECOMPOSITION

We start with introducing the components which will appear when upper-bounding (9). The primary quantity which we analyze in this section is the *error* between values of functional $\mathcal{L}$, i.e.:

$$\mathcal{L}(\varphi, T) - \mathcal{L}(\varphi', T'). \tag{10}$$

Depending on the context, the plug-in arguments $\varphi, \varphi', T, T'$ of the expression above may be "optimal" in some sense and connected to each other. In particular, the potentials $\varphi, \varphi'$ may solve the outer ($\min_\varphi$) optimization for the original (6) or empirical (7) objective in certain class of functions, i.e., $\mathcal{C}(\mathcal{Y})$ or $\mathcal{F}$. In turn, the maps $T, T'$ can be optimal in the inner ($\max_T$) optimization of the minimax objectives for certain potentials $\varphi$ and also belong to certain classes of functions. These choices give rise to several options, which are discussed below.

Typically, quantities (errors, bounds, etc.) appearing in statistical learning theory allow for decomposition into approximation and estimation components. We also treat error (10) in this framework.

**Approximation error.** This error arises because we are not optimizing over the full space of continuous functions $\varphi \in \mathcal{C}(\mathcal{Y})$ and measurable maps $T$, but over the restricted classes $\mathcal{F}$ and $\mathcal{T}$ of neural networks. Since neural networks have a limited capacity, they may not be able to represent the true saddle-point solution $(\varphi^*, T^*)$ of $\mathcal{L}(\varphi, T)$ exactly. We consider two components of this error, focusing respectively on the outer and inner optimization in (6).

The outer component assumes that the inner maximization ($\max_T$) is solved exactly. We want to identify to which extent the restricted outer minimization ($\min_{\varphi \in \mathcal{F}}$) affects the minimax objective:

$$\mathcal{E}_{Out}^A(\mathcal{F}) \overset{\text{def}}{=} \big| \min_{\varphi \in \mathcal{F}} \max_T \mathcal{L}(\varphi, T) - \min_\varphi \max_T \mathcal{L}(\varphi, T) \big|.$$

Note that $\max_T \mathcal{L}(\varphi, T) = \mathcal{L}(\varphi)$. Then the outer approximation error could be reformulated as:

$$\mathcal{E}_{Out}^A(\mathcal{F}) = \big| \min_{\varphi \in \mathcal{F}} \mathcal{L}(\varphi) - \min_\varphi \mathcal{L}(\varphi) \big|. \tag{11}$$

The inner approximation error, in turn, measures the looseness of inner maximization in the restricted class $\mathcal{T}$, i.e., the gap $| \max_T \mathcal{L}(\varphi, T) - \max_{T \in \mathcal{T}} \mathcal{L}(\varphi, T)|$. To have $\varphi$-independent characteristic, we further bound the gap uniformly w.r.t. $\varphi \in \mathcal{F}$. This yields

$$\mathcal{E}_{In}^A(\mathcal{F}, \mathcal{T}) = \max_{\varphi \in \mathcal{F}} \left| \max_T \mathcal{L}(\varphi, T) - \max_{T \in \mathcal{T}} \mathcal{L}(\varphi, T) \right|. \tag{12}$$

**Estimation error.** In practice, we do not have direct access to $\mathcal{L}(\varphi, T)$ but instead optimize the empirical functional $\widehat{\mathcal{L}}(\varphi, T)$ based on finite training data. Thus, the optimal potential and map which deliver optimal value to $\widehat{\mathcal{L}}$, i.e., $\widehat{\varphi}^R$ and $\widehat{T}^R$, may be different from those which deliver optimal value to $\mathcal{L}$, even within classes $\mathcal{F}, \mathcal{T}$. This introduces *estimation error*, which quantitatively measures the behaviour of functional $\mathcal{L}$ which takes "empirically-optimal" functions $\widehat{\varphi}^R, \widehat{T}^R$ as the input. By design, the estimation error is evaluated within classes $\mathcal{F}, \mathcal{T}$. Also, since $\widehat{\varphi}^R, \widehat{T}^R$ are (indirectly) based on the empirical samples $\{x_1, \ldots x_N\} = X \sim p$ and $\{y_1, \ldots, y_M\} = Y \sim q$, the estimation error is averaged w.r.t. them and depend on the sample sizes $N, M$.

The inner estimation error is defined as:

$$\mathcal{E}_{In}^E(\mathcal{F}, \mathcal{T}, N, M) \overset{\text{def}}{=} \mathbb{E}_{X,Y} \left| \max_{T \in \mathcal{T}} \mathcal{L}(\widehat{\varphi}^R, T) - \mathcal{L}(\widehat{\varphi}^R, \widehat{T}^R) \right|. \tag{13}$$

For the fixed potential $\widehat{\varphi}^R$ it shows how different are the optimal value of the inner ($\max_{T \in \mathcal{T}}$) optimization and the "empirically-induced" one $\mathcal{L}(\widehat{\varphi}^R, \widehat{T}^R)$.

Similarly, the outer estimation error deals with the outer minimization ($\min_{\varphi \in \mathcal{F}}$) while assuming exact solution of inner maximization in class $\mathcal{T}$:

$$\mathcal{E}_{Out}^E(\mathcal{F}, \mathcal{T}, N, M) = \mathbb{E}_{X,Y} \left| \max_{T \in \mathcal{T}} \mathcal{L}(\widehat{\varphi}^R, T) - \min_{\varphi \in \mathcal{F}} \max_{T \in \mathcal{T}} \mathcal{L}(\varphi, T) \right|. \tag{14}$$

Now we are ready to formulate our important Theorem 4.1 about the error decomposition.

**Theorem 4.1** (Error decomposition). *Let $\mathcal{F}$ be a class of $\beta$-strongly convex functions, then*

$$\mathbb{E}_{X,Y} \left\| \widehat{T}^R - T^* \right\|_{L_2(p)}^2 \leq \frac{4}{\beta} \left( \mathcal{E}_{In}^E(\mathcal{F}, \mathcal{T}, N, M) + 3\mathcal{E}_{In}^A(\mathcal{F}, \mathcal{T}) + \mathcal{E}_{Out}^E(\mathcal{F}, \mathcal{T}, N, M) + \mathcal{E}_{Out}^A(\mathcal{F}) \right). \tag{15}$$

The theorem shows that the (averaged) $L^2$ gap between the empirical and true OT maps (9) could be upper-bounded by our introduced errors. Importantly, the latter compares the values of functional $\mathcal{L}$ and thus easier for the analysis (conducted in the next subsection §4.2) than the former.

## 4.2 Bounds on the Approximation and Estimation Errors

In this section, we establish the bounds on the estimation and approximation errors of the minimax OT solvers defined in §4.1. In the theorem below, we establish the bounds on the total estimation error, i.e., the sum of inner (13) and outer errors (14). The bounds are given by Rademacher complexities of the certain classes of functions. Rademacher complexity is a standard tool in statistical learning theory; we recall its definition in Appendix A.2.

**Theorem 4.2** (Rademacher Bound on the Estimation Error). *Let $\mathcal{E}^E = \mathcal{E}^E_{In} + \mathcal{E}^E_{Out}$. Then*

$$\mathcal{E}^E \leq 8\mathcal{R}_{p,N}(\mathcal{H}) + 8\mathcal{R}_{q,M}(\mathcal{F}), \tag{16}$$

*where $\mathcal{H}(\mathcal{F}, \mathcal{T}) \stackrel{\text{def}}{=} \{h : h(x) = \langle x, T(x)\rangle - \varphi(T(x)), T \in \mathcal{T}, \varphi \in \mathcal{F}\}$ and $\mathcal{R}_{p,N}(\mathcal{H})$ is the Rademacher complexity of the function class $\mathcal{H}$ w.r.t. probability density $p$ for sample size $N$.*

Now we proceed to the approximation errors (12) and (11). In what follows, we prove that under proper choice of classes $\mathcal{F}, \mathcal{T}$ the errors could be done arbitrarily small.

First we look at the inner approximation error (12). Note that it depends on both classes $\mathcal{F}$ and $\mathcal{T}$. The following theorem states that by imposing certain restrictions on class $\mathcal{F}$ and by choosing an appropriate class of neural networks $\mathcal{T}$, we can control the inner approximation error.

**Theorem 4.3** (Inner approximation error). *Let $\mathcal{F} \subset \mathcal{C}(\mathcal{Y})$ be a class of Lipschitz $\beta$-strongly convex functions, totally bounded with respect to the Lipschitz norm. Then for all $\varepsilon > 0$ there exists a class of neural networks $\mathcal{T}_{ub} = \mathcal{T}(\varepsilon, \mathcal{F})$ and their Im-projected versions $\mathcal{T} = \{\text{proj}_{\mathcal{Y}} \circ T, T \in \mathcal{T}_{ub}\}$ s.t.:*

$$\mathcal{E}^A_{In}(\mathcal{F}, \mathcal{T}) < \varepsilon; \tag{17}$$

$\text{proj}_{\mathcal{Y}} : \mathbb{R}^D \to \mathcal{Y}$ *is the projection operator, see Lemma A.7 in Appendix A.3 for the formal definition.*

Importantly, the projection operator is not a restrictive add-on; it is usually used in practice, for example when working in the image domain (projection to pixel space). Our following Proposition 4.4 elaborates on the functional class $\mathcal{F}$ appearing in the theorem above.

**Proposition 4.4** (Class $\mathcal{F}$ in practice). *Let $\mathcal{F}_{icnn}$ be a class of K-layer ICNN with CELU activations, bounded width, height; bounded weights matrices $\|a_k\|_2 \leq A$, $k = 1, \ldots, K$; defined in proof of (Chen et al., 2018, Thm. 1). Then $\mathcal{F} = \{\varphi + \beta\frac{\|\cdot\|_2^2}{2}, \varphi \in \mathcal{F}_{icnn}\}$ satisfies the requirements of Theorem 4.3, i.e., $\mathcal{F}$ is a class of $\beta$-strongly convex functions totally bounded w.r.t. the Lipschitz norm.*

From the proof of Theorem 4.3 one can notice that the class of neural networks $\mathcal{T}$ for which (17) holds could be chosen to be finite. This naturally leads to another practically-important Remark 4.5:

*Remark* 4.5. In conditions of Theorem 4.3, the class of maps $\mathcal{T}$ could be chosen to be a ReLU Neural Network (MLP) with fixed width, height; with weights of bounded norms. Explicit capacities (e.g., Rademacher complexities, covering numbers, VC dimensions) of neural nets' classes are well-studied in the literature (Golowich et al., 2018; Ou & Bölcskei, 2024; Bartlett et al., 2019; Bach, 2017).

Now we turn to the analysis of the outer approximation error (11). Similarly to the inner case, under the assumption that the optimal potential $\varphi^*$ is $\beta$-strongly convex for some $\beta > 0$, we show that there exists Neural Network architecture for the dual potential $\varphi$, which makes the error arbitrarily small.

**Theorem 4.6** (Outer Approximation Error). *Let the optimal potential $\varphi^*$ be $\beta$ - strongly convex. For arbitrary $\varepsilon > 0$ there exists totally bounded w.r.t. Lipschitz norm class $\mathcal{F} = \mathcal{F}(\beta, \varepsilon)$ (does not depend on $\varphi^*$) such that for some $\beta$-strongly convex and Lipschitz function $\varphi^\beta_L$ it holds that $\varphi^\beta_L \in \mathcal{F}$ and $\mathcal{L}(\varphi^\beta_L) - \mathcal{L}(\varphi^*) < \varepsilon$.*

**Corollary 4.7.** *There exists a class of Lipschitz $\beta$-strongly convex functions, totally bounded with respect to the Lipschitz norm, such that*

$$\mathcal{E}^A_{Out}(\mathcal{F}) \leq \varepsilon. \tag{18}$$

In our Theorem 4.6 (and Corollary 4.7) we assume that the optimal potential $\varphi^*$ is $\beta$ - strongly convex. Our Remark 4.8 clarifies the conditions on the source and target distributions $p$ and $q$ which ensure the strong convexity.

*Remark* 4.8. Recall that the support subsets $\mathcal{X}$, $\mathcal{Y}$ be convex; let probability densities $p$ and $q$ be strictly positive and bounded on $\mathcal{X}$ and $\mathcal{Y}$, respectively; additionally, let probability densities $p, q$ be Hölder continuous functions, i.e., $p \in \mathcal{C}^\alpha(\mathcal{X}); q \in \mathcal{C}^\alpha(\mathcal{Y})$ for some $\alpha > 0$. Then, from (Caffarelli, 1996), Theorem ([C3]), it follows that optimal potential $\varphi^*$ is $\beta$-strongly convex for some $\beta > 0$.

It is noteworthy that the conditions on $p$ and $q$ are rather flexible; they are usually met for distributions that are commonly used in practice (with bounded densities) and for mixtures of distributions, for example mixtures of Gaussians (with the support artificially limited to a large enough ball $B_R(0), R \gg 1$, to ensure the compactness of $\mathcal{X}, \mathcal{Y}$).

### 4.3 BOUND ON THE GENERALIZATION ERROR

The main goal of our paper is the establishment of bounds on *generalization error*, i.e., the difference in true OT map and its empirical approximation defined in (9). In the theorem below, we use the previously obtained bounds on the estimation (Theorem 4.2) and approximation errors (Theorems 4.3, 4.6) to derive a bound on the generalization error.

**Theorem 4.9** (Bound on the Generalization Error). *Let the optimal dual potential $\varphi^*$ be $\beta$-strongly convex. Then for any $\varepsilon > 0$ there exist such classes $\mathcal{F} = \mathcal{F}(\varepsilon, \beta), \mathcal{T} = \mathcal{T}(\varepsilon, \mathcal{F})$ that*

$$\mathbb{E}_{X,Y} \left\| T^* - \widehat{T}^R \right\|_{L_2(p)}^2 \leq \varepsilon + \frac{32}{\beta} \left( \mathcal{R}_{p,N}(\mathcal{H}) + \mathcal{R}_{q,M}(\mathcal{F}) \right). \tag{19}$$

*Moreover, $\mathcal{F}$ and $\mathcal{T}$ could be chosen as neural networks following Proposition 4.4 and Remark 4.5.*

Theorem 4.9 shows that a practitioner can make the generalization error (9) arbitrarily small by choosing appropriate classes of functions $\mathcal{F}$ and $\mathcal{T}$. However, the numerical convergence rates remain unclear. The next corollary shows for the particular choice of functional classes (neural network), the obtained bound can be specified by replacing the Rademacher complexities with their upper bounds depending only on the number of empirical samples $(N, M)$.

**Corollary 4.10** (Generalization Error for the Specific Classes of Neural Networks). *For any $\varepsilon > 0$, let $\mathcal{T}, \mathcal{F}$ be classes of neural nets specified in Theorem 4.9. Then inequality (19) turns to*

$$\mathbb{E}_{X,Y} \left\| T^* - \widehat{T}^R \right\|_{L_2(p)}^2 \leq \varepsilon + O(\frac{1}{\sqrt{N}}) + O(\frac{1}{\sqrt{M}}). \tag{20}$$

The corollary allows us to conclude that the generalization error can be made arbitrarily small if one selects the appropriate classes of neural nets and sufficient number of empirical training samples.

## 5 EXPERIMENTAL ILLUSTRATIONS

In this section, we experimentally illustrate how the generalization error behaves in practice. Specifically, we show that the empirically observed generalization errors, i.e., the convergence rates of OT solver, are close to the bounds which we theoretically derive in our Corollary 4.10. For clarity of illustration, we study these rates for the examples where the generalization error is in the proximity of either the estimation error (§5.1) or the approximation error (§5.2). For both experiments, we utilize the benchmark from (Korotin et al., 2021b) which provides ground-truth OT maps $T^*$ and potentials $\varphi^*$ for the given source and target measures. In fact, the benchmark data *is constructed* based on the potential $\varphi^*$ parametrized by the specific NN architecture (ICNN). We consider their Gaussian mixture experiment, where the first distribution corresponds to a mixture of three Gaussians and the second to an *approximate* mixture of ten Gaussians. We refer to (Korotin et al., 2021b, Appendix A.1) for additional details on the experimental setup. Technical *training details* are given in Appendix D. The source code is written in PyTorch and is available online https://github.com/milenagazdieva/StatOT.

### 5.1 EXPERIMENT ON THE ESTIMATION ERROR

Below we study the effects of the estimation error coming from the limited number of data samples available for training. For this purpose, we use the same neural network architecture for potential $\varphi$ which was used for the ground-truth potential $\varphi^*$ in the benchmark. It allows us to assume that the outer approximation error (11) is arbitrarily small. Then, for fixed parametrization of OT maps $T_\omega$, the generalization error (19) reduces to estimation error (up to additive constant).

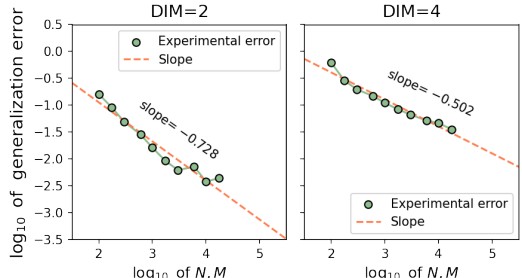

Figure 3: Convergence rates of the OT solver learned with the quadratic transport cost and a limited number of empirical training samples.

**Details.** We consider distributions of dimension $D = 2, 4$ and an OT solver with the quadratic cost $c(x, y) = \|x - y\|_2^2$. To show the dependence of the convergence rates on the number of available empirical samples $N, M$, we randomly select different numbers of samples (from $10^2$ to $2 \cdot 10^4$) from the source and target datasets, and train the OT solver using these samples. We set ICNN as the potential $\varphi_\theta$ and an MLP with ReLU activations

as the transport map $T_\omega$. During validation, we calculate the differences between the learned $\widehat{T}$ and true $T^*$ OT map $\|\widehat{T} - T^*\|^2_{L^2(p)}$ for all choices of $N,\ M$ and report them in Fig. 3. We plot the error using the logarithmic scale. Additionally, we run a linear regression on the logarithms of the error, as well as the logarithms of $N$, $M$. Then we plot the resulting line, which describes the relationship between these variables in $\log_{10}$-$\log_{10}$ space.

**Results** show that the logarithm of the error exhibits approximately linear convergence w.r.t. the logarithms of $N, M$ (with slope $\lesssim -0.5$). It supports our theoretically derived bounds on the error, see Corollary 4.10. Still, we note that the theoretical bounds may not hold in more complex practical setups due to the optimization errors coming from particular optimization procedures which are typically used to train the networks. In Appendix C.1, we report the estimation errors of several trivial _baselines._ Estimation errors of the OT solver for _higher dimensions_ are analyzed in Appendix C.2.

## 5.2 EXPERIMENT ON THE APPROXIMATION ERROR

In this section, we experimentally illustrate the effects of the approximation error coming from the limited expressivity of the architectures of neural nets $T_\omega$ and $\varphi_\theta$. For this purpose, we follow the experimental setup described in §5.1 with two important differences. First, we take $\approx 10M$ data samples from the benchmark, which allows us to assume that the estimation error is rather small and, thus, the generalization error (19) reduces to the approximation one. Second, we consider more shallow architectures for potential than the one proposed in the benchmark, which ensures that the approximation error does not vanish.

**Details.** We consider neural nets $\varphi_\theta$ of the same depth as in the architecture for the ground-truth potential $\varphi^*$, but with a limited size of hidden layers $H_\varphi$ ($\max H_\varphi$ from $4$ to $64$). Note that for $\varphi^*$, this value is equal to $64$. Besides, we vary the width of the NN architecture for the transport map $T$. We consider the architecture of the same depth as in our previous experiments, but with a varying size of hidden layers $\max H_T$ (from $1$ to $8$). In our previous experiments, we set $\max H_T = 512$. For each experiment, we calculate the difference between the learned and true OT maps: $\|\widehat{T} - T^*\|^2_{L^2(\mathbb{P})}$, as explained above. We plot the results using the logarithmic scale in Fig. 4.

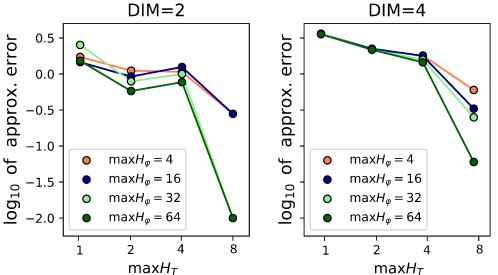

Figure 4: Empirical approximation error of the OT solver learned with the quadratic transport cost and using shallow NN architectures.

**Results** show the expected behavior − with the increase of the nets' width, the approximation error becomes smaller; when the potential architecture coincides with the one which was used during the benchmark construction ($\max H_\varphi = 64$), the approximation error is close to zero. Still, the trend might be corrupted by the optimization error, leading to outliers in the particular error values. Besides, the reader might notice that for nets' architectures with extremely small width, the approximation error might collapse to particular large values. Note that this is an expected behavior since the nets with too shallow architectures have very limited expressivity which might lead to improper solutions and large approximation errors. For clarity, we visualize the _solver's solutions_ in Appendix C.3.

## 6 DISCUSSION

Our paper performs theoretical investigation of semi-dual minimax OT solvers, i.e., a popular and fruitful branch of generative models based on Optimal Transport. While these solvers show impressive results in various machine learning problems, theoretical investigation of their practical convergence guarantees is rather limited. We address this problem by presenting the first theoretical analysis of these _minimax_ solvers from a _statistical learning_ perspective. Strictly speaking, some of the established results are more aligned with the _consistent estimation theory_ framework rather than statistical learning one. Most importantly, our paper provides _learnability guarantees_ for the solvers which justify their practical usability. Apart from the theoretical results, our paper provides experimental illustrations supporting the theoretical conclusions. We believe that our research will advance the development of minimax adversarial OT solvers by mitigating potential concerns regarding their theoretical validity.

*Limitations.* One limitation of our study corresponds to the focus on a popular case of a quadratic OT cost function. Besides, one can notice that our analysis employs quite strict $\beta$-strong convexity assumptions and does not provide lower bounds on the generalization error. Additionally, we do not provide practical guidance on selecting hyperparameters of neural nets parametrizing the map and potential in OT solvers. We discuss the *validity of the assumptions*, *lower bounds* and *practical relevance* of our analysis in Appendix B. Generalization of the established bounds for the general OT formulations represents a promising avenue for future work.

**Reproducibility.** A full list of the experimental details is provided in Appendix D. Our experiment uses synthetic dataset (benchmark) which is constructed via publicly available code of (Korotin et al., 2021b). We provide the code for the experiments in supplementary material.

## 7 ACKNOWLEDGMENTS

The work was supported by the grant for research centers in the field of AI provided by the Ministry of Economic Development of the Russian Federation in accordance with the agreement 000000C313925P4F0002 and the agreement №139-10-2025-033.

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

# A PROOFS.

In this section, we provide the proofs of our theoretical results:

- Theorem 4.1 (§A.1) which decomposes the recovered map $\widehat{T}^R$ error into several approximation and estimation subterms depending on the values of the optimized functional $\mathcal{L}$.
- Theorem 4.2 (§A.2) which upper-bounds the *estimation* error with Rademacher complexities of function classes $\mathcal{F}$, $\mathcal{H}$.
- Theorem 4.3 (§A.3) and Theorem 4.6 (§A.4) which demonstrate that the approximation error could be done arbitrary small under properly chosen classes of Neural Networks.
- Theorem 4.9 (§A.5) and Corollary 4.10 (§A.5) which establish upper bounds for the generalization error (9) and show that it can be made arbitrarily small by choosing the appropriate classes of functions and sufficient number of samples.

Before starting the main proofs, we state the following auxiliary lemma

**Lemma A.1.** *Let $A, B : \mathcal{G} \to \mathbb{R}$ be some functionals on a functional space $\mathcal{G}$. Then*

$$\left| \inf_{g \in \mathcal{G}} A(g) - \inf_{g \in \mathcal{G}} B(g) \right| \leq \sup_{g \in \mathcal{G}} |A(g) - B(g)|;$$

$$\left| \sup_{g \in \mathcal{G}} A(g) - \sup_{g \in \mathcal{G}} B(g) \right| \leq \sup_{g \in \mathcal{G}} |A(g) - B(g)|.$$

*Proof.* In what follows, we prove the first inequality in the statement of the lemma. The second inequality could be derived in a similar way.

Without loss of generality, assume that

$$\inf_g A(g) \geqslant \inf_g B(g) \implies \left| \inf_{g \in \mathcal{G}} A(g) - \inf_{g \in \mathcal{G}} B(g) \right| = \inf_{g \in \mathcal{G}} A(g) - \inf_{g \in \mathcal{G}} B(g).$$

Let us pick $\epsilon > 0$ and consider $g_\epsilon^* \in \mathcal{G}$ such that $B(g_\epsilon^*) \leq \inf_g B(g) + \epsilon$. Note that $\inf_g A(g) \leq A(g_\epsilon^*)$. Therefore:

$$\inf_{g \in \mathcal{G}} A(g) - \inf_{g \in \mathcal{G}} B(g) \leq A(g_\epsilon^*) - B(g_\epsilon^*) + \epsilon$$

$$\leq \left| A(g_\epsilon^*) - B(g_\epsilon^*) \right| + \epsilon$$

$$\leq \sup_{g \in \mathcal{G}} \left| A(g) - B(g) \right| + \epsilon.$$

Taking the limit $\epsilon \to 0$ in the last inequality finishes the proof. $\square$

## A.1 PROOF OF THEOREM 4.1

First, we upper-bound the error in estimating transport map via duality gaps analysis. Our theorem below borrows the main ideas from (Makkuva et al., 2020, Theorem 3.6), but has its own specificity, since it deals with transport maps that are not necessarily given by gradients of convex functions.

**Theorem A.2.** *Under the condition of theorem 4.1 it holds:*

$$\left\| \widehat{T}^R - T^* \right\|_{L^2(p)}^2 \leq \frac{4}{\beta} \left( \mathcal{E}_1(\widehat{\varphi}^R, \widehat{T}^R) + \mathcal{E}_2(\widehat{\varphi}^R) \right),$$

*where* $\mathcal{E}_1(\widehat{\varphi}^R, \widehat{T}^R) \overset{\text{def}}{=} \max_T \mathcal{L}(\widehat{\varphi}^R, T) - \mathcal{L}(\widehat{\varphi}^R, \widehat{T}^R)$ *is the inner error, and* $\mathcal{E}_2(\widehat{\varphi}^R) \overset{\text{def}}{=} \max_T \mathcal{L}(\widehat{\varphi}^R, T) - \min_\varphi \max_T \mathcal{L}(\varphi, T)$ *is the outer error.*

*Proof.* We introduce $T_{\widehat{\varphi}^R} \overset{\text{def}}{=} \arg\max_T \mathcal{L}(\widehat{\varphi}^R, T)$. Then,

$$\mathcal{E}_1(\widehat{\varphi}^R, \widehat{T}^R) = \int_{\mathcal{X}} \left[ \langle x, T_{\widehat{\varphi}^R}(x) \rangle - \widehat{\varphi}^R(T_{\widehat{\varphi}^R}(x)) \right] p(x)dx - \int_{\mathcal{X}} \left[ \langle x, \widehat{T}^R(x) \rangle - \widehat{\varphi}^R(\widehat{T}^R(x)) \right] p(x)dx.$$

Consider $\beta$-strongly concave $f_x(y) \overset{\text{def}}{=} \langle x, y \rangle - \widehat{\varphi}^R(y)$. Then,

$$\mathcal{E}_1(\widehat{\varphi}^R, \widehat{T}^R) = \int_{\mathcal{X}} \left[ f_x(T_{\widehat{\varphi}^R}(x)) - f_x(\widehat{T}^R(x)) \right] p(x) dx. \tag{21}$$

Note that $T_{\widehat{\varphi}^R}(x)$ is the maximizer of $f_x$ (cf. *Remark*). Thanks to $\beta$-strong concavity of $f_x$ it holds (for any $T$):

$$f_x(T_{\widehat{\varphi}^R}(x)) - f_x(T(x)) \geq \frac{\beta}{2} \left\| T(x) - T_{\widehat{\varphi}^R}(x) \right\|_2^2. \tag{22}$$

Combining (21) and (22), we get:

$$\mathcal{E}_1(\widehat{\varphi}^R, \widehat{T}^R) \geq \frac{\beta}{2} \left\| \widehat{T}^R - T_{\widehat{\varphi}^R} \right\|_{L^2(p)}^2. \tag{23}$$

Now we move to $\mathcal{E}_2(\widehat{\varphi}^R)$. We have the following:

$$\mathcal{E}_2(\widehat{\varphi}^R) = \mathcal{L}(\widehat{\varphi}^R, T_{\widehat{\varphi}^R}) - \mathcal{L}(\varphi^*, T^*)$$
$$= \int_{\mathcal{X}} \langle x, T_{\widehat{\varphi}^R}(x) \rangle p(x) dx - \int_{\mathcal{X}} \widehat{\varphi}^R(T_{\widehat{\varphi}^R}(x)) p(x) dx + \int_{\mathcal{Y}} \widehat{\varphi}^R(y) q(y) dy$$
$$- \int_{\mathcal{X}} \langle x, T^*(x) \rangle p(x) dx + \underbrace{\int_{\mathcal{X}} \varphi^*(T^*(x)) p(x) dx - \int_{\mathcal{Y}} \varphi^*(y) q(y) dy}_{=0, \text{ since } T^*_\# p = q}$$
$$= \int_{\mathcal{X}} \left[ \langle x, T_{\widehat{\varphi}^R}(x) \rangle - \widehat{\varphi}^R(T_{\widehat{\varphi}^R}(x)) \right] p(x) dx + \int_{\mathcal{Y}} \widehat{\varphi}^R(y) q(y) dy - \int_{\mathcal{X}} \langle x, T^*(x) \rangle p(x) dx \tag{24}$$
$$= \int_{\mathcal{X}} \left[ \langle x, T_{\widehat{\varphi}^R}(x) \rangle - \widehat{\varphi}^R(T_{\widehat{\varphi}^R}(x)) \right] p(x) dx + \int_{\mathcal{X}} \widehat{\varphi}^R(T^*(x)) p(x) dx - \int_{\mathcal{X}} \langle x, T^*(x) \rangle p(x) dx \tag{25}$$
$$= \int_{\mathcal{X}} \underbrace{\left[ \langle x, T_{\widehat{\varphi}^R}(x) \rangle - \widehat{\varphi}^R(T_{\widehat{\varphi}^R}(x)) \right]}_{=f_x(T_{\widehat{\varphi}^R}(x))} p(x) dx - \int_{\mathcal{X}} \underbrace{\left[ \langle x, T^*(x) \rangle - \widehat{\varphi}^R(T^*(x)) \right]}_{=f_x(T^*(x))} p(x) dx,$$
$$= \int_{\mathcal{X}} [f_x(T_{\widehat{\varphi}^R}(x)) - f_x(T^*(x))] p(x) dx \tag{26}$$
$$\geq \frac{\beta}{2} \int_{\mathcal{X}} \| T^*(x) - T_{\widehat{\varphi}^R}(x) \|_2^2 p(x) dx. \tag{27}$$

where in transition from (24) to (25) we use $T^*_\# p = q$ which yields $\int_{\mathcal{Y}} \widehat{\varphi}^R(y) q(y) dy = \int_{\mathcal{X}} \widehat{\varphi}^R(T^*(x)) p(x) dx$; in transition from (26) to (27) we apply property (22) for $T$ given by true OT map $T^*$.

Application of the triangle inequality for (23) and (27) finishes the proof:

$$\| T^* - \widehat{T}^R \|_{L_2(p)}^2 \overset{\triangle \text{ ineq.}}{\leq} \left( \| T^* - T_{\widehat{\varphi}^R} \|_{L_2(p)} + \| T_{\widehat{\varphi}^R} - \widehat{T}^R \|_{L_2(p)} \right)^2$$
$$\overset{\text{eqs. (23), (27)}}{\leq} \left( \sqrt{\frac{2}{\beta} \mathcal{E}_1(\widehat{\varphi}^R, \widehat{T}^R)} + \sqrt{\frac{2}{\beta} \mathcal{E}_2(\widehat{\varphi}^R)} \right)^2$$
$$\leq \frac{4}{\beta} \left( \mathcal{E}_1(\widehat{\varphi}^R, \widehat{T}^R) + \mathcal{E}_2(\widehat{\varphi}^R) \right).$$

$\square$

Now we proceed to the proof of the main theorem. To get the final result, it remains to decompose both $\mathcal{E}_1$ and $\mathcal{E}_2$ errors from the Theorem A.2 into estimation and approximation parts.

*Proof of Theorem 4.1.* From Theorem A.2 we have:

$$\mathop{\mathbb{E}}_{X,Y} \left\| \widehat{T}^R - T^* \right\|_{L^2(p)}^2 \leq \frac{4}{\beta} \left( \mathop{\mathbb{E}}_{X,Y} \mathcal{E}_1(\widehat{\varphi}^R, \widehat{T}^R) + \mathop{\mathbb{E}}_{X,Y} \mathcal{E}_2(\widehat{\varphi}^R) \right). \tag{28}$$

For the inner error $\mathcal{E}_1$ we have:

$$\mathcal{E}_1(\widehat{\varphi}^R, \widehat{T}^R) = \max_T \mathcal{L}(\widehat{\varphi}^R, T) - \mathcal{L}(\widehat{\varphi}^R, \widehat{T}^R)$$

$$\leq \left| \max_T \mathcal{L}(\widehat{\varphi}^R, T) - \max_{T \in \mathcal{T}} \mathcal{L}(\widehat{\varphi}^R, T) \right| + \left| \max_{T \in \mathcal{T}} \mathcal{L}(\widehat{\varphi}^R, T) - \mathcal{L}(\widehat{\varphi}^R, \widehat{T}^R) \right|$$

$$\leq \max_{\varphi \in \mathcal{F}} \left| \max_T \mathcal{L}(\varphi, T) - \max_{T \in \mathcal{T}} \mathcal{L}(\varphi, T) \right| + \left| \max_{T \in \mathcal{T}} \mathcal{L}(\widehat{\varphi}^R, T) - \mathcal{L}(\widehat{\varphi}^R, \widehat{T}^R) \right|.$$

Taking the expectation of both sides in the inequality above yields:

$$\mathop{\mathbb{E}}_{X,Y} \mathcal{E}_1(\widehat{\varphi}^R, \widehat{T}^R) \leq \mathcal{E}_{In}^A(\mathcal{F}, \mathcal{T}) + \mathcal{E}_{In}^E(\mathcal{F}, \mathcal{T}, N, M). \tag{29}$$

For the outer error $\mathcal{E}_2$ we have:

$$\mathcal{E}_2(\widehat{\varphi}^R) = \max_T \mathcal{L}(\widehat{\varphi}^R, T) - \min_\varphi \max_T \mathcal{L}(\varphi, T)$$

$$= \underbrace{\max_T \mathcal{L}(\widehat{\varphi}^R, T) - \max_{T \in \mathcal{T}} \mathcal{L}(\widehat{\varphi}^R, T)}_{\cdots} + \underbrace{\max_{T \in \mathcal{T}} \mathcal{L}(\widehat{\varphi}^R, T) - \min_{\varphi \in \mathcal{F}} \max_{T \in \mathcal{T}} \mathcal{L}(\varphi, T)}_{\sim}$$

$$+ \underbrace{\min_{\varphi \in \mathcal{F}} \max_{T \in \mathcal{T}} \mathcal{L}(\varphi, T) - \min_{\varphi \in \mathcal{F}} \max_T \mathcal{L}(\varphi, T)}_{----} + \underbrace{\min_{\varphi \in \mathcal{F}} \max_T \mathcal{L}(\varphi, T) - \min_\varphi \max_T \mathcal{L}(\varphi, T)}_{\_\_\_}$$

$$\tag{30}$$

$$\leq \underbrace{\max_{\varphi \in \mathcal{F}} \left| \max_{T \in \mathcal{T}} \mathcal{L}(\varphi, T) - \max_T \mathcal{L}(\varphi, T) \right|}_{\cdots} + \underbrace{\max_{T \in \mathcal{T}} \mathcal{L}(\widehat{\varphi}^R, T) - \min_{\varphi \in \mathcal{F}} \max_{T \in \mathcal{T}} \mathcal{L}(\varphi, T)}_{\sim}$$

$$+ \underbrace{\max_{\varphi \in \mathcal{F}} \left| \max_{T \in \mathcal{T}} \mathcal{L}(\varphi, T) - \max_T \mathcal{L}(\varphi, T) \right|}_{----} + \underbrace{\min_{\varphi \in \mathcal{F}} \mathcal{L}(\varphi) - \mathcal{L}(\varphi^*)}_{\_\_\_} \tag{31}$$

$$\leq \underbrace{\mathcal{E}_{In}^A(\mathcal{F}, \mathcal{T})}_{\cdots} + \underbrace{\left| \max_{T \in \mathcal{T}} \mathcal{L}(\widehat{\varphi}^R, T) - \min_{\varphi \in \mathcal{F}} \max_{T \in \mathcal{T}} \mathcal{L}(\varphi, T) \right|}_{\sim} + \underbrace{\mathcal{E}_{In}^A(\mathcal{F}, \mathcal{T})}_{----} + \underbrace{\mathcal{E}_{Out}^A(\mathcal{F})}_{\_\_\_}, \tag{32}$$

where in transition from (30) to (31) we apply Lemma A.1 for dashed terms.

Taking the expectation of the inequality (32) results in the following:

$$\mathop{\mathbb{E}}_{X,Y} \mathcal{E}_2(\widehat{\varphi}^R) \leq \mathcal{E}_{In}^A(\mathcal{F}, \mathcal{T}) + \mathcal{E}_{Out}^E(\mathcal{F}, \mathcal{T}, N, M) + \mathcal{E}_{In}^A(\mathcal{F}, \mathcal{T}) + \mathcal{E}_{Out}^A(\mathcal{F}). \tag{33}$$

The combination of (28), (29) and (33) finishes the proof. □

## A.2 PROOF OF THEOREM 4.2

Our theorem 4.2 uses some standard notions from learning theory, see, e.g. (Shalev-Shwartz & Ben-David, 2014, §26). We recall them for the convenience. For a class $\mathcal{F}$ of functions $\varphi : \mathcal{Y} \to \mathbb{R}$ and probability distribution $q$, the **representativeness** of a sample $Y = \{y_1, \ldots, y_M\} \sim q$ of size $M$:

$$\text{Rep}_{\mathcal{F},q}(Y) \overset{\text{def}}{=} \sup_{\varphi \in \mathcal{F}} \left| \int \varphi(y) q(y) dy - \frac{1}{M} \sum_{m=1}^M \varphi(y_m) \right|. \tag{34}$$

In turn, the **Rademacher complexity** of the class $\mathcal{F}$ w.r.t. the probability distribution $q$ and sample size $M$ is given by:

$$\mathcal{R}_{q,M}(\mathcal{F}) \overset{\text{def}}{=} \frac{1}{M} \mathop{\mathbb{E}}_{\sigma, Y} \left\{ \sup_{\varphi \in \mathcal{F}} \sum_{m=1}^M \varphi(y_m) \sigma_m \right\}, \tag{35}$$

where $Y = \{y_m\}_{m=1}^M \sim q$ are mutually independent, $\sigma = \{\sigma_m\}_{m=1}^M$ are mutually independent Rademacher random variables, i.e., $\text{Prob}(\sigma_m = 1) = \text{Prob}(\sigma_m = -1) = 0.5$, and the expectation is

taken with respect to both $Y$, $\sigma$. The quantities (35) and (34) have a well-known relation (Shalev-Shwartz & Ben-David, 2014, Lemma 26.2):

$$\mathbb{E}_Y \text{Rep}_{\mathcal{F},q}(Y) \leq 2\mathcal{R}_{q,M}(\mathcal{F}), \tag{36}$$

where the expectation is taken w.r.t. random i.i.d. sample $Y \sim q$ of size $M$.

Now we proceed to an auxiliary statement needed for the proof of our Theorem 4.2. In the next lemma, we upper-bound the difference between the original and sample-based functionals $\mathcal{L}$ and $\widehat{\mathcal{L}}$ with representativeness numbers w.r.t. classes $\mathcal{F}$, $\mathcal{H}$.

**Lemma A.3.** *Let $X$ and $Y$ be training samples from $p$ and $q$, respectively. Under the conditions of Theorem 4.2 it holds:*

$$\sup_{\varphi \in \mathcal{F}} \sup_{T \in \mathcal{T}} \left| \mathcal{L}(\varphi, T) - \widehat{\mathcal{L}}(\varphi, T) \right| \leq \text{Rep}_{\mathcal{H},p}(X) + \text{Rep}_{\mathcal{F},q}(Y), \tag{37}$$

*where $\text{Rep}_{\mathcal{H},p}(X)$ and $\text{Rep}_{\mathcal{F},q}(Y)$ are representativeness (34) of the corresponding samples w.r.t. corresponding classes.*

*Proof.* Let us write out in detail the right-hand-side of (37) and regroup the terms:

$$\sup_{\varphi \in \mathcal{F}} \sup_{T \in \mathcal{T}} \left| \mathcal{L}(\varphi, T) - \widehat{\mathcal{L}}(\varphi, T) \right| =$$

$$= \sup_{\varphi \in \mathcal{F}} \sup_{T \in \mathcal{T}} \left| \int \left[ \langle x, T(x) \rangle - \varphi(T(x)) \right] p(x) dx + \int \varphi(y) q(y) dy \right.$$

$$\left. - \frac{1}{N} \sum_{n=1}^{N} \left[ \langle x_n, T(x_n) \rangle - \varphi(T(x_n)) \right] - \frac{1}{M} \sum_{m=1}^{M} \varphi(y_m) \right|$$

$$\leq \sup_{\varphi \in \mathcal{F}} \sup_{T \in \mathcal{T}} \left\{ \left| \int \underbrace{\left[ \langle x, T(x) \rangle - \varphi(T(x)) \right]}_{=h(x), h \in \mathcal{H}} p(x) dx - \frac{1}{N} \sum_{n=1}^{N} \underbrace{\left[ \langle x_n, T(x_n) \rangle - \varphi(T(x_n)) \right]}_{=h(x_n)} \right| \right.$$

$$\left. + \left| \int \varphi(y) q(y) dy - \frac{1}{M} \sum_{m=1}^{M} \varphi(y_m) \right| \right\}$$

$$\leq \sup_{h \in \mathcal{H}} \left| \int h(x) p(x) dx - \frac{1}{N} \sum_{n=1}^{N} h(x_n) \right| + \sup_{\varphi \in \mathcal{F}} \left| \int \varphi(y) q(y) dy - \frac{1}{M} \sum_{m=1}^{M} \varphi(y_m) \right|$$

$$= \text{Rep}_{\mathcal{H},p}(X) + \text{Rep}_{\mathcal{F},q}(Y).$$

$\square$

Now we proceed to the proof of Theorem 4.2

*Proof of theorem 4.2.* We start by decomposing $\mathcal{E}_{In}^{E}$ and $\mathcal{E}_{Out}^{E}$ into subterms:

$$\mathcal{E}_{In}^{E}(\mathcal{F}, \mathcal{T}, N, M) = \mathbb{E}_{X,Y} \left| \mathcal{L}(\widehat{\varphi}^R, \widehat{T}^R) - \max_{T \in \mathcal{T}} \mathcal{L}(\widehat{\varphi}^R, T) \right|$$

$$\leq \mathbb{E}_{X,Y} \left[ \left| \mathcal{L}(\widehat{\varphi}^R, \widehat{T}^R) - \widehat{\mathcal{L}}(\widehat{\varphi}^R, \widehat{T}^R) \right| \right. \tag{38}$$

$$\left. + \left| \widehat{\mathcal{L}}(\widehat{\varphi}^R, \widehat{T}^R) - \max_{T \in \mathcal{T}} \mathcal{L}(\widehat{\varphi}^R, T) \right| \right]; \tag{39}$$

$$\mathcal{E}_{Out}^{E}(\mathcal{F}, \mathcal{T}, N, M) = \mathbb{E}_{X,Y} \left[ \max_{T \in \mathcal{T}} \mathcal{L}(\widehat{\varphi}^R, T) - \min_{\varphi \in \mathcal{F}} \max_{T \in \mathcal{T}} \mathcal{L}(\varphi, T) \right]$$

$$\leq \mathbb{E}_{X,Y} \left[ \left| \max_{T \in \mathcal{T}} \mathcal{L}(\widehat{\varphi}^R, T) - \widehat{\mathcal{L}}(\widehat{\varphi}^R, \widehat{T}^R) \right| \right. \tag{40}$$

$$\left. + \left| \widehat{\mathcal{L}}(\widehat{\varphi}^R, \widehat{T}^R) - \min_{\varphi \in \mathcal{F}} \max_{T \in \mathcal{T}} \mathcal{L}(\varphi, T) \right| \right]. \tag{41}$$

Now we analyze the subterms from the equations above separately. For simplicity, we omit the expectation " $\mathbb{E}_{X,Y}$ " sign.

Term (41)

$$\left| \widehat{\mathcal{L}}(\widehat{\varphi}^R, \widehat{T}^R) - \min_{\varphi \in \mathcal{F}} \max_{T \in \mathcal{T}} \mathcal{L}(\varphi, T) \right| = \left| \min_{\varphi \in \mathcal{F}} \max_{T \in \mathcal{T}} \widehat{\mathcal{L}}(\varphi, T) - \min_{\varphi \in \mathcal{F}} \max_{T \in \mathcal{T}} \mathcal{L}(\varphi, T) \right|. \qquad (42)$$

Notice that $\max_{T \in \mathcal{T}} \mathcal{L}(\varphi, T)$ and $\max_{T \in \mathcal{T}} \widehat{\mathcal{L}}(\varphi, T)$ are some functionals, depending on $\varphi$ (denote them as $l(\varphi)$ and $\widehat{l}(\varphi)$). Then we can apply Lemma A.1:

$$\left| \min_{\varphi \in \mathcal{F}} l(\varphi) - \min_{\varphi \in \mathcal{F}} \widehat{l}(\varphi) \right| \leq \max_{\varphi \in \mathcal{F}} \left| l(\varphi) - \widehat{l}(\varphi) \right| = \max_{\varphi \in \mathcal{F}} \left| \max_{T \in \mathcal{T}} \widehat{\mathcal{L}}(\varphi, T) - \max_{T \in \mathcal{T}} \mathcal{L}(\varphi, T) \right|. \qquad (43)$$

Applying Lemma A.1 again to the internal expression, we get:

$$\max_{\varphi \in \mathcal{F}} \left| \max_{T \in \mathcal{T}} \widehat{\mathcal{L}}(\varphi, T) - \max_{T \in \mathcal{T}} \mathcal{L}(\varphi, T) \right| \leq \max_{\varphi \in \mathcal{F}} \max_{T \in \mathcal{T}} \left| \widehat{\mathcal{L}}(\varphi, T) - \mathcal{L}(\varphi, T) \right|. \qquad (44)$$

Terms (39), (40).

Similarly, we apply Lemma A.1:

$$\left| \widehat{\mathcal{L}}(\widehat{\varphi}^R, \widehat{T}^R) - \max_{T \in \mathcal{T}} \mathcal{L}(\widehat{\varphi}^R, T) \right| = \left| \max_{T \in \mathcal{T}} \widehat{\mathcal{L}}(\widehat{\varphi}^R, T) - \max_{T \in \mathcal{T}} \mathcal{L}(\widehat{\varphi}^R, T) \right|$$

$$\leq \max_{T \in \mathcal{T}} \left| \widehat{\mathcal{L}}(\widehat{\varphi}^R, T) - \mathcal{L}(\widehat{\varphi}^R, T) \right|. \qquad (45)$$

Finally, we get that all four terms (38), (39), (40), (41) are bounded by $\max_{\varphi \in \mathcal{F}} \max_{T \in \mathcal{T}} \left| \widehat{\mathcal{L}}(\varphi, T) - \mathcal{L}(\varphi, T) \right|$. Therefore, by applying Lemma A.3 we get:

$$\mathcal{E}_{In}^E(\mathcal{F}, \mathcal{T}, N, M) + \mathcal{E}_{Out}^E(\mathcal{F}, \mathcal{T}, N, M) \leq \mathbb{E}_{X,Y} \left[ 4 \max_{\varphi \in \mathcal{F}} \max_{T \in \mathcal{T}} \left| \widehat{\mathcal{L}}(\varphi, T) - \mathcal{L}(\varphi, T) \right| \right]$$

$$\leq 4 \mathbb{E}_X \operatorname{Rep}_{\mathcal{H},p}(X) + 4 \mathbb{E}_Y \operatorname{Rep}_{\mathcal{F},q}(Y)$$

$$\overset{(36)}{\leq} 8 \mathcal{R}_{p,N}(\mathcal{H}) + 8 \mathcal{R}_{q,M}(\mathcal{F}),$$

which finishes the proof.

$\square$

### A.3 PROOF OF THEOREM 4.3

To begin with, we recall some standard definitions and notions used in this subsection. Given function $\varphi \in \mathcal{C}(\mathcal{Y})$, its **Lipschitz (semi)norm** is defined as:

$$\|\varphi\|_{Lip} \overset{\text{def}}{=} \sup_{\substack{y \neq y' \\ y, y' \in \mathcal{Y}}} \frac{|\varphi(y) - \varphi(y')|}{\|y - y'\|_2}.$$

(Continuous) functions with bounded Lipschitz norm are called **Lipschitz continuous** and denoted as $Lip(\mathcal{Y}) \subset \mathcal{C}(\mathcal{Y})$.

Now we proceed to the main content of the subsection. The next auxiliary lemma states that the differences in the functional $\mathcal{L}$ values could be controlled by the properties of its arguments.

**Lemma A.4.** *Let $\varphi \in Lip(\mathcal{Y})$ and $T_1, T_2 : \mathcal{X} \to \mathcal{Y}$ be measurable mappings. Then,*

$$|\mathcal{L}(\varphi, T_1) - \mathcal{L}(\varphi, T_2)| \leq \left( \sup_{x \in \mathcal{X}} \|x\|_2 + \|\varphi\|_{Lip} \right) \|T_1 - T_2\|_{L_1(p)}.$$

*Proof.* Consider the left-hand-side of the desired inequality:

$$\left| \mathcal{L}(\varphi, T_1) - \mathcal{L}(\varphi, T_2) \right| = \left| \int_{\mathcal{X}} \left\{ \left( \langle x, T_1(x) \rangle - \varphi(T_1(x)) \right) - \left( \langle x, T_2(x) \rangle - \varphi(T_2(x)) \right) \right\} p(x) dx \right|$$

$$\overset{\text{Jensen ineq.}}{\leq} \int_{\mathcal{X}} \left| \left( \langle x, T_1(x) \rangle - \varphi(T_1(x)) \right) - \left( \langle x, T_2(x) \rangle - \varphi(T_2(x)) \right) \right| p(x) dx.$$
(46)

Consider the expression under integral sign:

$$\left| \left( \langle x, T_1(x) \rangle - \varphi(T_1(x)) \right) - \left( \langle (x, T_2(x) \rangle - \varphi(T_2(x)) \right) \right|$$

$$\leq \left| \langle x, T_1(x) \rangle - \langle x, T_2(x) \rangle \right| + \left| \varphi(T_1(x)) - \varphi(T_2(x)) \right|$$

$$\leq \|x\|_2 \|T_1(x) - T_2(x)\|_2 + \|\varphi\|_{Lip} \|T_1(x) - T_2(x)\|_2. \quad (47)$$

The substitution of (47) into (46) finishes the proof. □

**Corollary A.5.** *Let* $T_\varphi^R = \arg\max_{T \in \mathcal{T}} \mathcal{L}(\varphi, T)$, $T_\varphi = \arg\max_T \mathcal{L}(\varphi, T)$. *Then,*

$$\max_T \mathcal{L}(\varphi, T) - \max_{T \in \mathcal{T}} \mathcal{L}(\varphi, T) \leq \left( \sup_{x \in \mathcal{X}} \|x\|_2 + \|\varphi\|_{Lip} \right) \left\| T_\varphi^R - T_\varphi \right\|_{L_1(p)}.$$

The next auxiliary lemma upper-bounds the $L_1(p)$ norm between maps which are optimal for given dual potentials.

**Lemma A.6.** *Let* $\varphi_1, \varphi_2 \in Lip(\mathcal{Y})$ *be* $\beta$-*strongly convex. Then:*

$$\|T_{\varphi_1} - T_{\varphi_2}\|_{L_1(p)} \leq \frac{1}{\beta} \|\varphi_1 - \varphi_2\|_{Lip},$$

*where* $T_{\varphi_1} = \arg\max_T \mathcal{L}(\varphi_1, T)$; $T_{\varphi_2} = \arg\max_T \mathcal{L}(\varphi_2, T)$.

*Proof.* We pick $x \in \mathcal{X}$ and consider function $y \mapsto \langle x, y \rangle - \varphi_1(y)$. Then:

- $T_{\varphi_1}(x)$ is the maximizer of $y \mapsto \langle x, y \rangle - \varphi_1(y)$ (see *Remark*);
- $y \mapsto \langle x, y \rangle - \varphi_1(y)$ is a $\beta$-strongly concave function.

By the property of strongly-concave function, we have:

$$\langle x, T_{\varphi_1}(x) \rangle - \varphi_1(T_{\varphi_1}(x)) - \frac{\beta}{2} \|T_{\varphi_1}(x) - T_{\varphi_2}(x)\|_2^2 \geq \langle x, T_{\varphi_2}(x) \rangle - \varphi_1(T_{\varphi_2}(x)) \Longrightarrow$$

$$\frac{\beta}{2} \|T_{\varphi_1}(x) - T_{\varphi_2}(x)\|_2^2 \leq \varphi_1(T_{\varphi_2}(x)) - \varphi_1(T_{\varphi_1}(x)) + \langle x, T_{\varphi_1}(x) \rangle - \langle x, T_{\varphi_2}(x) \rangle.$$
(48)

Similar inequality holds for $\varphi_2$:

$$\langle x, T_{\varphi_2}(x) \rangle - \varphi_2(T_{\varphi_2}(x)) - \frac{\beta}{2} \|T_{\varphi_1}(x) - T_{\varphi_2}(x)\|_2^2 \geq \langle x, T_{\varphi_1}(x) \rangle - \varphi_2(T_{\varphi_1}(x)) \Longrightarrow$$

$$\frac{\beta}{2} \|T_{\varphi_1}(x) - T_{\varphi_2}(x)\|_2^2 \leq \varphi_2(T_{\varphi_1}(x)) - \varphi_2(T_{\varphi_2}(x)) + \langle x, T_{\varphi_2}(x) \rangle - \langle x, T_{\varphi_1}(x) \rangle.$$
(49)

Combining inequalities (48) and (49) we get:

$$\beta \|T_{\varphi_1}(x) - T_{\varphi_2}(x)\|_2^2 \leq [\varphi_1 - \varphi_2](T_{\varphi_2}(x)) - [\varphi_1 - \varphi_2](T_{\varphi_1}(x))$$

$$\leq \|\varphi_1 - \varphi_2\|_{Lip} \|T_{\varphi_1}(x) - T_{\varphi_2}(x)\|_2,$$

i.e., $\|T_{\varphi_1}(x) - T_{\varphi_2}(x)\|_2 \leq \frac{1}{\beta} \|\varphi_1 - \varphi_2\|_{Lip}$. Taking the expectation w.r.t. probability distribution $p(x)$ completes the proof. □

The next lemma is a technical result, which allows to restrict the output of mappings $T^{NN} : \mathcal{X} \to \mathbb{R}^D$ parameterized with neural networks to compact support space $\mathcal{Y}$ with help of a *projection* function.

**Lemma A.7.** *Consider the projection operator* $\text{proj}_{\mathcal{Y}} : \mathbb{R}^D \to \mathcal{Y}$, *defined as :*

$$\text{proj}_{\mathcal{Y}}(r) = \underset{y \in \mathcal{Y}}{\arg\min} \|y - r\|_2.$$

*Then,* $\forall r_1, r_2 \in \mathbb{R}^D$ *it holds:*

$$\|\text{proj}_{\mathcal{Y}}(r_1) - \text{proj}_{\mathcal{Y}}(r_2)\|_2 \leq \|r_1 - r_2\|_2. \tag{50}$$

*In particular,* $\text{proj}_{\mathcal{Y}}$ *is a 1-Lipschitz function.*

*Proof.* To begin with, we prove the following technical fact.

**Part I. Technical fact**. Let $y \in \mathcal{Y}$, $r \in \mathbb{R}^D \backslash \mathcal{Y}$. Let $y_\star = \text{proj}_{\mathcal{Y}}(r)$. Then:

$$\langle y - y_\star, y_\star - r \rangle \geq 0. \tag{51}$$

Consider variable $\beta \in [0, 1]$, and $y_\beta = (1 - \beta)y_\star + \beta y$. Note that $y_\beta \in \mathcal{Y}$, since $\mathcal{Y}$ is convex. We have the following:

$$\|y_\beta - r\|_2^2 = \|y_\beta - y_\star + y_\star - r\|_2^2$$
$$= \beta^2 \|y - y_\star\|_2^2 + 2\beta \langle y - y_\star, y_\star - r \rangle + \|y_\star - r\|_2^2.$$

By definition of $\text{proj}_{\mathcal{Y}}$ it holds: $\|y_\beta - r\|_2 \geq \|y_\star - r\|_2$. Then,

$$\beta^2 \|y - y_\star\|_2^2 + 2\beta \langle y - y_\star, y_\star - r \rangle \geq 0;$$
$$2\beta \langle y - y_\star, y_\star - r \rangle \geq -\beta^2 \|y - y_\star\|_2^2 \quad \text{(divide by } 2\beta > 0\text{)};$$
$$\langle y - y_\star, y_\star - r \rangle \geq -\frac{\beta}{2} \|y - y_\star\|_2^2.$$

The last inequality holds for every $\beta > 0$. Therefore, (51) holds true.

Now, we move on to proving the statement of the Lemma. We consider different options:

(i) $r_1, r_2 \in \mathcal{Y}$;

(ii) $r_1 \in \mathcal{Y}$ and $r_2 \in \mathbb{R}^D \backslash \mathcal{Y}$ (equivalent to $r_2 \in \mathcal{Y}$ and $r_1 \in \mathbb{R}^D \backslash \mathcal{Y}$);

(iii) $r_1, r_2 \in \mathbb{R}^D \backslash \mathcal{Y}$.

Note that case (i) is trivial. We proceed to cases (ii) and (iii).

**Part II. Proof of the Lemma; case (ii).**

For ease of perception, introduce $y_1 := \text{proj}_{\mathcal{Y}}(r_1) = r_1 \in \mathcal{Y}$. Let $y_{\star,2} = \text{proj}_{\mathcal{Y}}(r_2)$. Then:

$$\|r_1 - r_2\|_2^2 = \|y_1 - r_2\|_2^2 = \|y_1 - y_{\star,2} + y_{\star,2} - r_2\|_2^2$$
$$= \underbrace{\|y_{\star,2} - r_2\|_2^2}_{\geq 0} + \underbrace{\|y_1 - y_{\star,2}\|_2^2}_{=\|\text{proj}_{\mathcal{Y}}(r_1) - \text{proj}_{\mathcal{Y}}(r_2)\|_2^2} + 2\underbrace{\langle y_1 - y_{\star,2}, y_{\star,2} - r_2 \rangle}_{\geq 0, \text{ see } (51)}$$
$$\geq \|\text{proj}_{\mathcal{Y}}(r_1) - \text{proj}_{\mathcal{Y}}(r_2)\|_2^2$$

**Part III. Proof of the Lemma; case (iii).**

Let $y_{\star,1} = \text{proj}_{\mathcal{Y}}(r_1)$; $y_{\star,2} = \text{proj}_{\mathcal{Y}}(r_2)$. We have the following:

$$\|r_1 - r_2\|_2^2 = \|r_1 - y_{\star,1} + y_{\star,1} - y_{\star,2} + y_{\star,2} - r_2\|_2^2$$
$$= \underbrace{\|r_1 - y_{\star,1} + y_{\star,2} - r_2\|_2^2}_{\geq 0} + \underbrace{\|y_{\star,1} - y_{\star,2}\|_2^2}_{=\|\text{proj}_{\mathcal{Y}}(r_1) - \text{proj}_{\mathcal{Y}}(r_2)\|_2^2}$$
$$+ 2\underbrace{\langle r_1 - y_{\star,1}, y_{\star,1} - y_{\star,2} \rangle}_{\geq 0, \text{ see } (51)} + 2\underbrace{\langle y_{\star,2} - r_2, y_{\star,1} - y_{\star,2} \rangle}_{\geq 0, \text{ see } (51)}$$
$$\geq \|\text{proj}_{\mathcal{Y}}(r_1) - \text{proj}_{\mathcal{Y}}(r_2)\|_2^2$$

$\square$

Now we proceed to the proof of the main theorem statement.

*Proof of theorem 4.3.* First notice, that from compactness of $\mathcal{F}$ it follows that $\mathcal{F}$ is uniformly bounded, i.e, $\exists L \in \mathbb{R} : \forall \varphi \in \mathcal{F} \, \|\varphi\|_{Lip} \leq L$.

For a fixed arbitrary small $\varepsilon$ pick $\delta_1 = \frac{\beta\varepsilon}{2\left(\sup_{x\in\mathcal{X}} \|x\|+L\right)}$. From the definition of compactness, it follows that there exists a finite $\delta_1$-covering of $\mathcal{F}$:

$$\forall \delta_1 > 0 \; \exists I = I(\delta_1), \; \exists \varphi_1, \ldots, \varphi_n \in \mathcal{F} : \; \mathcal{F} \subset \bigcup_{i=1}^{I} \mathcal{F}_i, \tag{52}$$

where $\mathcal{F}_i = \{\varphi : \; \varphi \in \mathcal{F}, \|\varphi - \varphi_i\|_{Lip} \leq \delta_1\}$

Pick $\delta_2 = \frac{\varepsilon}{2\left(\sup_{x\in\mathcal{X}} \|x\|+L\right)}$. According to (Kidger & Lyons, 2020, Theorem 4.16) for each $T_{\varphi_k}$ there exists neural network $T_{\varphi_k}^{NN(\delta_2)}$, such that $\left\|T_{\varphi_k}^{NN(\delta_2)} - T_{\varphi_k}\right\|_{L_1(p)} \leq \delta_2$ for arbitrary small $\delta_2$. Thanks to Lemma A.7, $\left\|\mathrm{proj}_{\mathcal{Y}} \circ T_{\varphi_k}^{NN(\delta_2)} - T_{\varphi_k}\right\|_{L_1(p)} \leq \left\|T_{\varphi_k}^{NN(\delta_2)} - T_{\varphi_k}\right\|_{L_1(p)} \leq \delta_2$. Let's choose class of neural networks $\mathcal{T}_{ub}(\delta_2)$ such that it includes all the functions $T_{\varphi_1}^{NN(\delta_2)}, \ldots, T_{\varphi_n}^{NN(\delta_2)}$. Then for such class:

$$\max_{T} \mathcal{L}(\varphi_k, T) - \max_{T\in\mathcal{T}_{ub}(\delta_2)} \mathcal{L}(\varphi_k, \mathrm{proj}_{\mathcal{Y}} \circ T) \leq \max_{T} \mathcal{L}(\varphi_k, T) - \mathcal{L}(\varphi_k, \mathrm{proj}_{\mathcal{Y}} \circ T_{\varphi_k}^{NN(\delta_2)}). \tag{53}$$

Now we are ready to bound inner approximation error.

Consider $\varphi \in \mathcal{F}$. Then $\varphi$ belongs to one of the sets of a finite coverage $\mathcal{F}_k$. According to lemma A.6 we have $\|T_{\varphi_k} - T_{\varphi}\|_{L_1(p)} \leq \frac{1}{\beta}\|\varphi_k - \varphi\|_{Lip} \leq \frac{\delta_1}{\beta}$

We also can approximate $T_{\varphi_k}$ by the neural network, so we can do it with $T_{\varphi}$:

$$\left\|\mathrm{proj}_{\mathcal{Y}} \circ T_{\varphi_k}^{NN(\delta_2)} - T_{\varphi}\right\|_{L_1(p)} \leq \left\|\mathrm{proj}_{\mathcal{Y}} \circ T_{\varphi_k}^{NN(\delta_2)} - T_{\varphi_k}\right\|_{L_1(p)} + \|T_{\varphi_k} - T_{\varphi}\|_{L_1(p)} \leq \delta_2 + \frac{\delta_1}{\beta}. \tag{54}$$

Then it remains to apply Lemma A.4 and use the uniform boundness of $\mathcal{F}$:

$$\max_{T} \mathcal{L}(\varphi, T) - \mathcal{L}(\varphi, \mathrm{proj}_{\mathcal{Y}} \circ T_{\varphi_k}^{NN(\delta_2)}) \leq \left(\sup_{x\in\mathcal{X}} \|x\| + \|\varphi\|_{Lip}\right) \left\|\mathrm{proj}_{\mathcal{Y}} \circ T_{\varphi}^{NN(\delta_2)} - T_{\varphi}\right\|_{L_1(p)}$$

$$\leq \left(\sup_{x\in\mathcal{X}} \|x\| + L\right)\left(\delta_2 + \frac{\delta_1}{\beta}\right) = \varepsilon. \tag{55}$$

Due to the arbitrariness of the choice of $\varphi$, we obtain

$$\max_{\varphi\in\mathcal{F}} \left[\max_{T} \mathcal{L}(\varphi, T) - \max_{T\in\mathcal{T}_{ub}(\delta_2)} \mathcal{L}(\varphi, \mathrm{proj}_{\mathcal{Y}} \circ T)\right] \leq \varepsilon \tag{56}$$

$\square$

### A.3.1 PROOF OF PROPOSITION 4.4

Let us denote $\varphi_s$ as a $K$-layer ICNN (Amos et al., 2017) with skip connections of the form

$$z_1(y) = s(a_1 y + b_1)$$
$$\ldots$$
$$z_{i+1}(y) = s(w_{i+1} z_i(y) + a_{i+1} y + b_{i+1}) \tag{57}$$
$$\ldots$$
$$z_K(y) = s(w_K z_{K-1}(y) + a_K y + b_K),$$

where $w_i \in W_i$, $a_i \in A_i$, $b_i \in B_i$, and for $i$-th layer $W_i$ ($W_1 = 0$), $A_i$ and $B_i$ are compact subsets of $\mathbb{R}_+^{D_{i+1} \times D_i}$ (nonnegative weights), $\mathbb{R}^{D_{i+1} \times D}$ and $\mathbb{R}^{D_{i+1}}$ respectively for any $i$-th layer, $s$ is a convex and non-decreasing activation function; $D_{K+1} = 1$ (ICNNs have scalar output). If $s$ is ReLU, then $\varphi_s$ is a neural network from universal approximation theorem for Lipschitz convex functions (Chen et al., 2018, Theorem 1). Further for convenience we denote $h_i = w_i z_{i-1}(y) + a_i y + b_i$; $p(\varphi_s) = \{(w_i, a_i, b_i) | i = 1, \ldots, K\}$; $\mathcal{P} = \{(w_i, a_i, b_i) | w_i \in W_i, a_i \in A_i, b_i \in B_i\}_{i=1}^K$ is the whole parameters space; $\mathcal{P}_n = \{w_1, a_1, b_1, \ldots, w_K, a_K, b_K\}$ are parameters' "names".

**Lemma A.8.** *Let $s$ be a smooth function. Let $p \in \mathcal{P}_n$ be a parameter of ICNN $\varphi_s$. Then $\forall \delta > 0$ $\exists \varepsilon > 0$ such that $\|p - p^1\|_2 \le \varepsilon \Rightarrow \|\varphi_s - \varphi_s^1\|_{Lip} < \delta$.*

*Proof.* Notice that $\varphi_s$ is differentiable. Then $\|\varphi_s\|_{Lip} = \max_{y \in \mathcal{Y}} \|\nabla \varphi_s(y)\|_2$.

Consider $\xi(y, p) \stackrel{\text{def}}{=} \nabla \varphi_{s|p=p}(y)$, i.e., $\xi$ is the gradient (w.r.t to input $y$) of ICNN $\varphi_s$ with specific value of parameter $p$. In particular, $\xi(y, p^1) = \nabla \varphi_s^1(y)$. Note that $\varphi_s$ is smooth (as the composition of smooth functions), both w.r.t input $y$ and parameters $\mathcal{P}$. Therefore, $\xi(y, p)$ is continuous. To finish the proof we are left to note that $\xi(y, p)$ is uniformly continuous ($y, p$ are from compact sets). $\square$

**Lemma A.9.** (ICNNs (57) are totally bounded). *Let $\mathcal{F}$ be a class of $K$-layer ICNNs of the form 57 with smooth activation $s$. Then $\mathcal{F}$ is totally bounded with respect to the Lipschitz norm.*

*Proof.* Pick arbitrary $\delta > 0$. From Lemma A.8, for each parameter $p \in \mathcal{P}_n$, we can choose $\varepsilon_p > 0$ such that:

$$\|p - p^1\|_2 < \varepsilon_p \Rightarrow \|\varphi_s - \varphi_s^1\|_{Lip} < \frac{\delta}{|\mathcal{P}_n|},$$

where $|\mathcal{P}_n|$ is the number of parameters in the neural network (it is finite). Define $\varepsilon := \min_{p \in \mathcal{P}_n} \varepsilon_p$.

Each parameter $p$ belongs to a compact set $P$; we consider a (finite) $\varepsilon$-net $\mathcal{E}_p$ that covers $P$. Then, the Cartesian product

$$\mathcal{E} \stackrel{\text{def}}{=} \bigotimes_{p \in \mathcal{P}_n} \mathcal{E}_p$$

defines the (finite) $\varepsilon$-net that covers the whole parameters set $\mathcal{P}$. Within each "ball" $\varsigma \in \mathcal{E}$, different combinations of parameters $(p_1^\varsigma, p_2^\varsigma, \ldots, p_{|\mathcal{P}|}^\varsigma), (p_1^{\varsigma'}, p_2^{\varsigma'}, \ldots, p_{|\mathcal{P}|}^{\varsigma'}) \in \varsigma$ are $\varepsilon$-close to each other point-wise, i.e., $\|p_i^\varsigma - p_i^{\varsigma'}\|_2 \le \varepsilon$. By subsequently substituting $p_i^\varsigma$ with $p_i^{\varsigma'}$, $i = 1, 2, \ldots, |\mathcal{P}_n|$ we obtain:

$$\|\varphi_s^\varsigma - \varphi_s^{\varsigma'}\|_{Lip} < \delta,$$

i.e., $\mathcal{E}$ defines $\delta$-net for $\varphi_s$. Thus, $\mathcal{F}$ is totally bounded with respect to the Lipschitz norm. $\square$

## A.4 PROOF OF THEOREM 4.6

Below, we will use the notion of $\mathcal{B}$-smoothness. Function $\varphi : \mathcal{Y} \to \mathbb{R}$ is called $\mathcal{B}$-**smooth** if it is continuously differentiable on $\mathcal{Y}$, and its gradient is Lipschitz continuous with Lipschitz constant $\mathcal{B}$:

$$\|\nabla \varphi(y_1) - \nabla \varphi(y_2)\|_2 \le \mathcal{B} \|y_1 - y_2\|_2, \quad \forall y_1, y_2 \in \mathcal{Y}.$$

We proceed with several auxiliary results needed for the main proof.

**Lemma A.10.** *Let $\varphi$ is $\mathcal{B}$-smooth on $\mathcal{Y}$. Then there exists $L > 0$ such that $\varphi$ is $L$-Lipschitz, i.e., $\|\varphi\|_{Lip} \le L$.*

*Proof.* By fundamental theorem of calculus along a segment $y$ to $y'$ we have:

$$\varphi(y) - \varphi(y') = \int_0^1 \langle \nabla \varphi(y' + t(y - y')), y - y' \rangle \, dt.$$

Applying Cauchy-Schwarz inequality we get:

$$|\varphi(y) - \varphi(y')| \le \int_0^1 \|\nabla\varphi(y' + t(y - y'))\|_2 \, \|y - y'\|_2 \, dt. \tag{58}$$

Given that $\nabla\varphi$ is continuous (because $\varphi$ is $\mathcal{B}$-smooth) and $\mathcal{Y}$ is bounded, then there exists $L$ defined as $L = \sup_{y \in \mathcal{Y}} \|\nabla\varphi(y)\|_2 < \infty$. Replacing in (58) we get

$$|\varphi(y) - \varphi(y')| \le L \|y - y'\|_2 \,,$$

which concludes the proof. $\qquad\square$

**Lemma A.11** (Property of convex conjugate w.r.t. sup norm). *Let $\varphi_1, \varphi_2 \in \mathcal{C}(\mathcal{Y})$. Then,*

$$\|\overline{\varphi_1} - \overline{\varphi_2}\|_\infty \le \|\varphi_1 - \varphi_2\|_\infty.$$

*If $\varphi_1$ and $\varphi_2$ are convex, then the inequality becomes an equality, i.e., $\|\overline{\varphi_1} - \overline{\varphi_2}\|_\infty = \|\varphi_1 - \varphi_2\|_\infty$.*

*Proof.* We pick $x \in \mathcal{X}$. Then,

$$
\begin{aligned}
|\overline{\varphi_1}(x) - \overline{\varphi_2}(x)| \quad &= \quad |\max_{y \in \mathcal{Y}}[\langle x, y\rangle - \varphi_1(y)] - \max_{y \in \mathcal{Y}}[\langle x, y\rangle - \varphi_2(y)]| \\
&\overset{\text{Lemma A.1}}{\le} \quad \max_{y \in \mathcal{Y}} |\langle x, y\rangle - \varphi_1(y) - \langle x, y\rangle + \varphi_2(y)| \\
&\le \quad \|\varphi_1 - \varphi_2\|_\infty.
\end{aligned}
$$

Taking the supremum w.r.t. $x$ in the inequality above yields $\|\overline{\varphi_1} - \overline{\varphi_2}\|_\infty \le \|\varphi_1 - \varphi_2\|_\infty$.

If $\varphi_1$ and $\varphi_2$ are convex, then:

$$\|\varphi_1 - \varphi_2\|_\infty = \|\overline{\overline{\varphi_1}} - \overline{\overline{\varphi_2}}\|_\infty \le \|\overline{\varphi_1} - \overline{\varphi_2}\|_\infty,$$

which proves the remaining statement. $\qquad\square$

**Lemma A.12.** *Let $\Phi$ be a set of convex functions on compact domain $\mathcal{Y} \subset \mathbb{R}^D$ and uniformly bounded with respect to the Lipschitz norm. Then for any $\varepsilon > 0$ there exists such constant $L = L(\varepsilon)$, that any $\varphi \in \Phi$ can be arbitrarily closely approximated by an $L$-Lipschitz function, i.e., for any $\varepsilon > 0$ one can choose a Lipschitz continuous function $\varphi_L \in Lip(\mathcal{Y})$, $\|\varphi_L\|_{Lip} \le L$ such that*

$$\|\varphi - \varphi_L\|_\infty \le \varepsilon.$$

*Proof.* Consider $\varphi \in \Phi$ and $\varepsilon > 0$ and take $\mathcal{B} = \mathcal{B}(\varepsilon) = \frac{\sup_{\mathcal{Y}} \|y\|_2^2}{2\varepsilon}$. Consider the following functions (recall that $\overline{\varphi}$ is convex conjugate of $\varphi$):

$$\psi(x) \overset{\text{def}}{=} \overline{\varphi}(x) + \frac{\|x\|_2^2}{2\mathcal{B}} \quad ; \quad \varphi_L \overset{\text{def}}{=} \overline{\psi} = \overline{\overline{\varphi} + \frac{\|\cdot\|_2^2}{2\mathcal{B}}}.$$

Note that $\psi$ is $\frac{1}{\mathcal{B}}$-strongly convex. Therefore, by (Kakade et al., 2009, Theorem 6) $\varphi_L$ is $\mathcal{B}$-smooth. Applying Lemma A.10 we get that $\varphi_L$ is also Lipschitz with some constant $L = L(\varepsilon)$ (note that $L$ does not depend on particular $\varphi$ thanks to uniform boundedness of $\Phi$ w.r.t. lischitz norm).

Finally,

$$\|\varphi - \varphi_L\|_\infty \overset{\text{Lemma A.11}}{=} \|\overline{\varphi} - \overline{\varphi_L}\|_\infty = \left\|\overline{\varphi} - \overline{\varphi} + \frac{\|\cdot\|_2^2}{2L_s}\right\|_\infty = \sup_{\mathcal{Y}} \frac{\|y\|_2^2}{2\mathcal{B}} = \varepsilon,$$

which completes the proof. $\qquad\square$

The next Lemma improves on (Chen et al., 2018, Lemma 1).

**Lemma A.13.** *Let $\Phi$ be a set of convex functions on compact domain $\mathcal{Y} \subset \mathbb{R}^D$ and uniformly bounded w.r.t. the Lipschitz and sup norms. Then for each $\varepsilon > 0$ one can choose $K$ such that $\forall \varphi \in \Phi$ it can be approximated by maximum of at most $K$ of affine functions with parameters on compact sets. That is $\forall \varphi \in \Phi$ there exists $\widehat{\varphi}(y) = \max_{i=1,\dots,n}\{\mu_i^\top y + b_i\}$, such that $\|\varphi - \widehat{\varphi}\|_\infty < \varepsilon$ and $n \le K$, $\mu_i \in M, b_i \in B$, where $M \subset \mathbb{R}^D$ and $B \subset \mathbb{R}$ are compact (and do not depend on particular $\varphi$).*

*Proof.* Note that $\mathcal{Y}$ is compact. Then there exists finite $\frac{\varepsilon}{4L}$-net $\mathcal{E}$ of $\mathcal{Y}$, $|\mathcal{E}| = K$. In what follows we approximate convex function $\varphi$ with linear function on each of the $\frac{\varepsilon}{4L}$-balls.

Note that $\Phi$ is uniformly bounded, and there exists the global Lipschitz constant $L$: $\|\varphi\|_{Lip} \leq L$.

For each $\frac{\varepsilon}{4L}$-ball $\varsigma_i \in \mathcal{E}$ let us choose an interior point $y_i^1$, where $\varphi$ is differentiable and consider tangent plane $\mu_i^\top y + b_i$ at this point. Note that $\varphi(y_i^1) = \mu_i^\top y_i^1 + b_i$ and $\forall y \in \mathcal{Y}: \varphi(y) \geq \mu_i^\top y + b_i$.

These planes are also $L$-Lipschitz, therefore for any other point $y_i^2 \in \varsigma_i$ it holds:

$$|\varphi(y_i^2) - \varphi(y_i^1)| \leq L \left\| y_i^2 - y_i^1 \right\| \leq \frac{\varepsilon}{2};$$

$$|(\mu_i^\top y_i^2 + b_i) - (\mu_i^\top y_i^1 + b_i)| \leq L \left\| y_i^2 - y_i^1 \right\| \leq \frac{\varepsilon}{2}.$$

Summing up these inequalities, we get:

$$\left| \varphi(y_i^2) - (\mu_i^\top y_i^2 + b_i) - \underbrace{\left[ \varphi(y_i^1) - (\mu_i^\top y_i^1 + b_i) \right]}_{=0} \right| \leq \varepsilon,$$

i.e., on ball $\varsigma_i$ our function $\varphi$ could be approximated from below by the corresponding plane $\mu_i^\top y + b_i$.

Now we define $\widehat{\varphi}(y) \stackrel{\text{def}}{=} \max_{i=1,\ldots,K}\{\mu_i^\top y + b_i\} : \mathcal{Y} \to \mathbb{R}$, where $(\mu_i, b_i)$ correspond to each $\varsigma_i$. Obviously,

$$|\varphi(y) - \widehat{\varphi}(y)| = \varphi(y) - \widehat{\varphi}(y) \leq \varepsilon, \forall y \in \mathcal{Y}.$$

In remains to notice that by property of Lipschitz functions we have $\|\mu_i\|_2 \leq L$. In turn, the compactness of parameters $b_i$ follows from the uniform boundedness of $\Phi$ w.r.t. sup norm ($\| \cdot \|_\infty$). $\square$

Taking the advantage of our Lemma A.13, we can formulate an improvement over (Chen et al., 2018, Theorem 1):

**Corollary A.14.** *For any $\varepsilon > 0, L > 0, C > 0$ we can choose $K$ and compact parameters set $\mathcal{P}$ such that for any convex function $\varphi$ with $\|\varphi\| \leq C, \|\varphi\|_{Lip} \leq L$ defined over a compact domain $\mathcal{Y}$ there exists an ICNN (57) $\varphi_\sigma$ with depth and width limited by $K$, with weights $p \in \mathcal{P}$ and ReLU activation functions $\sigma$ that approximates it within $\varepsilon$:*

$$\|\varphi - \varphi_\sigma\|_\infty < \varepsilon.$$

*Proof.* Follows the proof of (Chen et al., 2018, Thm. 1), but utilizes Lemma (A.13) instead of (Chen et al., 2018, Lemma 1). $\square$

**Lemma A.15.** *Let $s_n \rightrightarrows \sigma$ on any bounded set. Then $\forall \varepsilon > 0$ and parameters space $\mathcal{P}$ we can choose $n = n(\varepsilon, \mathcal{P}) \in \mathbb{N}$ such that two ICNNs (57) $\varphi_{s_n}$ and $\varphi_\sigma$ with coinciding weights $p(\varphi_{s_n}) = p(\varphi_\sigma)$ and different activation functions $s_n$ and $\sigma$ respectively are close by $\infty$-norm:*

$$\|\varphi_{s_n} - \varphi_\sigma\|_\infty < \varepsilon.$$

*In particular, $s_n$ could be chosen to be $CELU_n$ activation functions $CELU_n(x) = \max(0, x) + \min\left(0, \frac{1}{n}(e^{nx} - 1)\right)$, $\sigma$ is ReLU activation.*

*Proof.* First, let's notice that for fixed compact $\mathcal{P}$ from 57, compact input domain and bounded activation functions, output of each layer in ICNN belongs to a compact set, i.e.

$$\max_{p(\varphi) \in \mathcal{P}, y \in \mathcal{Y}} \|\varphi(y)\|_\infty < \text{Const}$$

Let's denote layers of two neural networks $\varphi_{s_n}$ and $\varphi_\sigma$ as $z_k^{s_n}$ and $z_k^\sigma$. We prove the statement by induction. Consider the difference of the first layer outputs':

$$\|z_1^{s_n}(y) - z_1^\sigma(y)\|_\infty = \|s_n(h_1^{s_n}(y)) - \sigma(h_1^\sigma(y))\|_\infty$$

As weights are equal, then $h_1^{s_n}(y) = h_1^\sigma(y)$. Recalling that for $p \in \mathcal{P}$ function (of parameters and inputs) $h_1^{s_n}(y)$ takes values over a compact set, due to uniform convergence of $s_n$, we have

$$\max_{p(\varphi) \in \mathcal{P}, y \in \mathcal{Y}} \|z_1^{s_n}(y) - z_1^\sigma(y)\|_\infty \to 0$$

Consider then difference of $k$-th layers

$$\|z_k^{s_n}(y) - z_k^\sigma(y)\|_\infty = \|s_n(h_k^{s_n}(y)) - \sigma(h_k^\sigma(y))\|_\infty \le$$
$$\|s_n(h_k^{s_n}(y)) - s_n(h_k^\sigma(y))\|_\infty + \|s_n(h_k^\sigma(y)) - \sigma(h_k^\sigma(y))\|_\infty. \quad (59)$$

Consider the first term of (59). Note that $\|h_k^{s_n}(y) - h_k^\sigma(y)\|_\infty = \|w_k(z_{k-1}^{s_n}(y) - z_{k-1}^\sigma(y))\|_\infty$. Therefore, due to continuity of $s_n$ and boundedness of $w_k \in W_k$ it holds:

$$\max_{p \in \mathcal{P}, x \in \mathcal{X}} \|s_n(y_k^{s_n}(x)) - s_n(y_k^\sigma(x))\|_\infty \to 0.$$

The second term $\|s_n(y_k^\sigma(x)) - \sigma(y_k^\sigma(x))\|_\infty$ uniformly tends to zero because of assumption of our lemma $s_n \rightrightarrows \sigma$.

Finally we get that $\max_{p \in P, x \in \mathcal{X}} \|z_k^{s_n}(x) - z_k^\sigma(x)\|_\infty \to 0$ for $k$-th layer. And, therefore by induction, we have uniform convergence for the ICNNs itself:

$$\max_{p \in \mathcal{P}, y \in \mathcal{Y}} \|\varphi_{s_n}(y) - \varphi_\sigma(y)\|_\infty \to 0.$$

$\square$

**Lemma A.16.** *Let $\Phi$ be a class of convex functions, uniformly bounded w.r.t. Lipschitz and* $\sup$ *norms. Then for a fixed $\varepsilon > 0$ there exists a class $\mathcal{F}$ of ICNNs (57) with $CELU_1$ activation totally bounded with respect to the Lipschitz norm, such that*

$$\forall \varphi \in \Phi \, \exists \varphi_s \in \mathcal{F} : \|\varphi - \varphi_s\|_\infty < \varepsilon.$$

*Proof.* By Corollary A.14 we can choose ICNN $\varphi_\sigma$ of the form 57 with width and depth bounded by $K = K(\varepsilon)$ and weights from compact set of parameters $\mathcal{P} = \mathcal{P}(\varepsilon)$ approximating $\varphi$ within $\frac{\varepsilon}{2}$:

$$\|\varphi_\sigma - \varphi\|_\infty < \frac{\varepsilon}{2}.$$

By Lemma A.15 we can choose $n = n(\varepsilon, \mathcal{P})$ such that replacing ReLU activation by $CELU_n$ will lead to controllable changes in $\varphi_\sigma$:

$$\|\varphi_\sigma - \varphi_{s_n}\|_\infty < \frac{\varepsilon}{2}.$$

Summing up the inequalities above we obtain:

$$\|\varphi_{s_n} - \varphi\|_\infty < \varepsilon.$$

Note that $CELU_n(x) = \frac{1}{n}CELU_1(nx)$. Therefore, $CELU_n$ activation functions in $\varphi_{s_n}$ could be substituted with $CELU_1$ by proper modification of parameters set $\mathcal{P}$ (which does not change the compactness of $\mathcal{P}$). $\square$

Now we are ready to prove the main Theorem.

*Proof of theorem 4.6.* By the assumption of the theorem $\varphi^* = \arg \min_\varphi \mathcal{L}(\varphi)$ is $\beta$-strongly convex. Consider

$$\varphi^{*, -\beta} \stackrel{\text{def}}{=} \varphi^* - \beta \frac{\|\cdot\|_2^2}{2}.$$

Note that $\varphi^{*, -\beta}$ is convex. From Lemma A.12 , we can find $\varphi_L \in Lip(\mathcal{Y})$ such that $\|\varphi_L\|_{Lip} \le L = L(\varepsilon)$ and:

$$\|\varphi^{*, -\beta} - \varphi_L\|_\infty < \frac{\varepsilon}{4}.$$

According to A.16 there exists class $\mathcal{F}_{icnn} = \mathcal{F}_{icnn}(\varepsilon)$ such that we can find there neural network $\varphi_L^{nn}$ close to any $L$-Lipschitz function:

$$\|\varphi_L - \varphi_L^{nn}\|_\infty < \frac{\varepsilon}{4}. \tag{60}$$

Obviously, $\varphi_L^{nn}$ is also Lipschitz since it is a composition of piecewise linear functions. Consider

$$\varphi_L^\beta \overset{\text{def}}{=} \varphi_L^{nn} + \beta \frac{\|\cdot\|_2^2}{2}.$$

Note that $\varphi_L^\beta$ is $\beta$-strongly convex. Then,

$$\|\varphi^* - \varphi_L^\beta\|_\infty = \|\varphi^{*,-\beta} - \varphi_L^{nn}\|_\infty \leq \|\varphi^{*,-\beta} - \varphi_L\|_\infty + \|\varphi_L - \varphi_L^{nn}\|_\infty \leq \frac{\varepsilon}{2}.$$

Let us check that $\varphi_L^\beta$ delivers the desired bound.

$$\mathcal{L}(\varphi_L^\beta) - \mathcal{L}(\varphi^*) = \int_{\mathcal{X}} \overline{\varphi_L^\beta}(x)p(x)dx + \int_{\mathcal{Y}} \varphi_L^\beta(y)q(y)dy - \left( \int_{\mathcal{X}} \overline{\varphi^*}(x)p(x)dx + \int_{\mathcal{Y}} \varphi^*(y)q(y)dy \right)$$

$$\leq \int_{\mathcal{X}} \left| \overline{\varphi^*}(x) - \overline{\varphi_L^\beta}(x) \right| p(x)dx + \int_{\mathcal{Y}} \left| \varphi^*(y) - \varphi_L^\beta(y) \right| q(y)dy$$

$$\leq \int_{\mathcal{X}} \underbrace{\left\| \overline{\varphi^*} - \overline{\varphi_L^\beta} \right\|_\infty}_{=\left\| \varphi^* - \varphi_L^\beta \right\|_\infty \text{ by Lemma A.11}} p(x)dx + \int_{\mathcal{Y}} \left\| \varphi^* - \varphi_L^\beta \right\|_\infty q(y)dy$$

$$= \varepsilon,$$

Therefore class $\mathcal{F} = \{\varphi + \beta \frac{\|\cdot\|_2^2}{2}, \varphi \in \mathcal{F}_{icnn}\}$ satisfies the condition of the theorem. $\qquad\square$

*Proof of corollary 4.7.* It remains to provide some class $\mathcal{F}$, which contains $\varphi_L^\beta$. We can choose sufficiently expressive class $\mathcal{F}_{icnn}$ and then define $\mathcal{F}$ such that $\varphi \in \mathcal{F}$ has the form $x \mapsto \varphi_\theta(x) + \beta \frac{\|x\|_2^2}{2}$ where $\varphi_\theta \in \mathcal{F}_{icnn}$. For compactness of $\mathcal{F}$ by Lipschitz norm we can bound all weight matrices by second norm. $\qquad\square$

### A.5 PROOF OF THEOREM 4.9 AND COROLLARY 4.10

*Proof of theorem 4.9.* Let's bound all the terms from decomposition in theorem 4.1 step by step. From corollary 4.7 we get such class $\mathcal{F} = \mathcal{F}(\frac{\beta\varepsilon}{8}, \beta)$ that $\mathcal{E}_{Out}^A(\mathcal{F}) < \frac{\beta\varepsilon}{8}$. The second step is to apply theorem 4.3, which gives us $\mathcal{T} = \mathcal{T}(\frac{\beta\varepsilon}{24}, \mathcal{F})$, such that $3\mathcal{E}_{In}^A(\mathcal{F}, \mathcal{T}) < \frac{\beta\varepsilon}{8}$. Finally, applying theorem 4.2, we bound remaining two terms

$$\mathcal{E}_{In}^E(\mathcal{F}, \mathcal{T}, N, M) + \mathcal{E}_{Out}^E(\mathcal{F}, \mathcal{T}, N, M) \leq 8\mathcal{R}_{p,N}(\mathcal{H}) + 8\mathcal{R}_{q,M}(\mathcal{F}).$$

$\qquad\square$

*Proof of corollary 4.10.* Our goal is to obtain bounds depending on sample sizes for Rademacher complexities of function classes $\mathcal{H} = \mathcal{H}(\mathcal{F}, \mathcal{T})$ and $\mathcal{F}$ which appear in the inequality (19) of Theorem 4.9. Recall that: $\mathcal{T} = \{\text{proj}_{\mathcal{Y}} \circ T, T \in \mathcal{T}_{ub}\}$; $\mathcal{T}_{ub}$ is a class of feedforward neural network with bounded norms of weights, limited width and height and Lipschitz activation functions $-$ ReLU; every function $\varphi \in \mathcal{F}$ has the form $x \mapsto \varphi_\theta(x) + \beta \frac{\|x\|_2^2}{2}$ where $\varphi_\theta$ belongs to the class of input convex neural networks $\mathcal{F}_{icnn}$. Here we additionally assume that neural nets from $\mathcal{F}_{icnn}$ have bounded norms and CELU activation functions. Let $\mathcal{T}'$ be a more general class of feedforward neural networks with any Lipschitz activation functions $\sigma(x)$, e.g., ReLU, CELU and others, s.t. $\sigma(0) = 0$. Note that the Rademacher complexity of class $\mathcal{T}'$ is known to attain an upper bound: $\mathcal{R}_{p,N} \leq O(\frac{1}{\sqrt{N}})$ where $O(\cdot)$ hides the constants which do not depend on the sample size $N$, see, e.g., (Golowich et al., 2018).

*Step 1.* We start with obtaining an upper bound for $\mathcal{R}_{q,M}(\mathcal{F})$. From the well-known monotonicity property of Rademacher complexity, we get that

$$\mathcal{F}_{icnn} \subset \mathcal{T}' \implies \mathcal{R}_{q,M}(\mathcal{F}_{icnn}) \le \mathcal{R}_{q,M}(\mathcal{T}') \le O(\frac{1}{\sqrt{M}}).$$

The complexity of the constrained quadratic functions $x \mapsto \frac{\beta}{2}\|x\|_2^2$ is also $O(\frac{1}{\sqrt{M}})$. It follows from their representation using the Reproducing Kernel Hilbert Spaces (RKHS), see, e.g., (Mohri et al., 2018, Thm. 6.12), (Latorre et al., 2021, Lemma 5, Eq. 24). Then, from the additivity of Rademacher complexities it follows that the complexity of functional class $\mathcal{F}$ is also bounded by $O(\frac{1}{\sqrt{M}})$.

*Step 2.* Below we prove that $\mathcal{R}_{p,N}(\mathcal{H})$ attains an upper bound $O(\frac{1}{\sqrt{N}})$. Recall that class $\mathcal{H}$ consists of functions $h$ which map $x \mapsto \langle x, \text{proj}_{\mathcal{Y}}(T(x)) \rangle - \varphi(\text{proj}_{\mathcal{Y}}(T(x)))$ where $T \in \mathcal{T}_{ub} \subset \mathcal{T}', \varphi \in \mathcal{F} \subset \mathcal{T}'$. Note that the function $\varphi \circ \text{proj}_{\mathcal{Y}} \circ T$ is a composition of feedforward neural networks with the Lipschitz activations from $\mathcal{T}'$ (plus additional intermediate 1-Lipschitz "activation" $\text{proj}_{\mathcal{Y}}$, see Lemma A.7), and, thus, belongs to the same class $\mathcal{T}'$. By monotonocity property, the complexity of this class of functions is bounded by $O(\frac{1}{\sqrt{N}})$ which follows from the first step of this proof.

It remains to show that the complexity of the class of functions $x \mapsto \langle x, \text{proj}_{\mathcal{Y}}(T(x)) \rangle$ for $T \in \mathcal{T}_{ub}$ is also bounded by $O(\frac{1}{\sqrt{N}})$. To prove this fact, we consider **empirical** Rademacher complexity for this class of functions ($\mathcal{T}''$) defined as

$$\widehat{\mathcal{R}}_{p,N}(\mathcal{T}'') \overset{\text{def}}{=} \frac{1}{N}\mathbb{E}_\sigma\left\{\sup_{f \in \mathcal{T}''}\sum_{n=1}^N f(x_n)\sigma_n\right\} = \frac{1}{N}\mathbb{E}_\sigma\left\{\sup_{T \in \mathcal{T}}\sum_{n=1}^N \langle x_n, \text{proj}_{\mathcal{Y}}(T(x_n))\rangle\sigma_n\right\}. \quad (61)$$

Here the difference with the classic Rademacher complexity definition (35) consists in the fact that the expectation is taken only w.r.t. the Rademacher random variables $\sigma = \{\sigma_n\}_{n=1}^N$. Importantly, we can derive (35) from (61) by taking the additional expectation w.r.t. random samples $X = \{x_1, ..., x_n\} \sim p$.

For simplicity of further derivations, we introduce function $T' : \mathbb{R}^D \to \mathbb{R}^{2D}$ s.t. $T'(x) = (x, T(x))$. Besides, we consider function $h : \mathcal{X} \times \mathbb{R}^D \mapsto \mathbb{R}$ s.t. $h(x, r) \overset{\text{def}}{=} \langle x, \text{proj}_{\mathcal{Y}}(r) \rangle$. Then (61) can be rewritten as:

$$\widehat{\mathcal{R}}_{p,N}(\mathcal{T}'') = \frac{1}{N}\mathbb{E}_{\sigma \sim \{\pm 1\}^N}\left\{\sup_{T' \in \mathcal{T}''}\sum_{n=1}^N (h \circ T')(x_n)\sigma_n\right\}. \quad (62)$$

We start with proving thet $h$ is a $G$-Lipschitz function. Indeed, let $x_1, x_2$ and $r_1, r_2$ belong to $\mathcal{X}$, $\mathbb{R}^D$, respectively. Let $y_1 = \text{proj}_{\mathcal{Y}}(r_1), y_2 = \text{proj}_{\mathcal{Y}}(r_2)$. Recall that $\mathcal{X}, \mathcal{Y}$ are compact subsets of $\mathbb{R}^D$ which means that for all $x \in \mathcal{X}, y \in \mathcal{Y}$ they have bounded norms: $\max(\|x\|_2, \|y\|_2) \le R$. Then:

$$|h(x_1, r_1) - h(x_2, r_2)|^2 = |\langle x_1, y_1\rangle - \langle x_2, y_2\rangle|^2 = |\langle x_1, y_1 - y_2\rangle + \langle x_1 - x_2, y_2\rangle|^2 \le$$
$$|\langle x_1, y_1 - y_2\rangle|^2 + |\langle x_1 - x_2, y_2\rangle|^2 \le \|x_1\|_2^2 \cdot \|y_1 - y_2\|_2^2 + \|y_1\|_2^2 \cdot \|x_1 - x_2\|_2^2 \le$$
$$R^2 \cdot (\|y_1 - y_2\|_2^2 + \|x_1 - x_2\|_2^2) \overset{\text{cf. (50)}}{\le} R^2 \cdot (\|r_1 - r_2\|_2^2 + \|x_1 - x_2\|_2^2) = \underbrace{R^2}_{G \overset{\text{def}}{=}}\|(x_1, y_1) - (x_2, y_2)\|_2^2.$$

This allows us to apply (Lei et al., 2023, Lemma B.1) (taken from (Maurer, 2016)) to get

$$(62) \overset{(Maurer, 2016)}{\le} \frac{1}{N}\sqrt{2}G \mathbb{E}_{\sigma \sim \{\pm 1\}^{2Nd}}\left\{\sup_{T' \in \mathcal{T}''}\sum_{n=1}^N\sum_{t=1}^{2d} T'_t(x_n)\sigma_{n,t}\right\} \le$$

$$\frac{1}{N}\sqrt{2}G \mathbb{E}_{\sigma \sim \{\pm 1\}^{Nd}}\left\{\sum_{n=1}^N\sum_{t=1}^d \underbrace{[x_n]_t}_{=T'_t(x_n)}\sigma_{n,t}\right\} + \frac{1}{N}\sqrt{2}G \mathbb{E}_{\sigma \sim \{\pm 1\}^{Nd}}\left\{\sup_{T \in \mathcal{T}}\sum_{n=1}^N\sum_{t=1}^d \underbrace{T_t(x_n)}_{=T'_{t+d}(x_n)}\sigma_{n,t}\right\}. \quad (63)$$

Applying (Lei et al., 2023, Proposition 5.5), we get that

$$\mathbb{E}_{\sigma \sim \{\pm 1\}^{Nd}}\left\{\sup_{T \in \mathcal{T}}\sum_{n=1}^N\sum_{t=1}^d T_t(x_n)\sigma_{n,t}\right\} \le \sqrt{d}C_1 \cdot \sqrt{C_2 \cdot (\sum_{1 \le i < j \le N}(x_i^T x_j)^2)^{\frac{1}{2}} + \sum_{j=1}^N \|x_j\|_2^2} \quad (64)$$

where $C_1$, $C_2$ are two constants which depend on only the number of layers in neural nets from $\mathcal{T}_{ub}$ and bounds on the weights. Besides,

$$\underset{\sigma \sim \{\pm 1\}^{Nd}}{\mathbb{E}} \sum_{n=1}^{N} \sum_{t=1}^{d} [x_n]_t \sigma_{n,t} = \sum_{n=1}^{N} \sum_{t=1}^{d} [x_n]_t \underset{\sigma \sim \{\pm 1\}^{Nd}}{\mathbb{E}} \overbrace{\sigma_{n,t}}^{0} = 0$$

Using these observations, we get

$$(63) \leq \frac{1}{N} \sqrt{2} G \left\{ \sqrt{d} C_1 \sqrt{C_2 \cdot (\sum_{1 \leq i < j \leq N} (x_i^T x_j)^2)^{\frac{1}{2}} + \sum_{j=1}^{N} \|x_j\|_2^2} \right\}. \tag{65}$$

In remains to take expectation of (65) w.r.t. $X = \{x_1, ..., x_N\} \sim p$. Before moving forward, we note that $\{x_1, ..., x_N\} \sim p$ are i.i.d. vectors.

Additionally note that since we are working on the compact subsets of $\mathbb{R}^D$ for each $i \neq j$:

$$\underset{X}{\mathbb{E}}(x_i^T x_j)^2 \leq \underset{X}{\mathbb{E}}\|x_i\|_2^2 \cdot \underset{X}{\mathbb{E}}\|x_j\|_2^2 \leq R^4.$$

At the same time, for each $i$ we have $\underset{X}{\mathbb{E}}\|x_i\|_2^2 \leq R^2$. Then we get,

$$\frac{1}{N} \sqrt{2d} G C_1 \cdot \underset{X}{\mathbb{E}} \left\{ \sqrt{C_2 \cdot (\sum_{1 \leq i < j \leq N} (x_i^T x_j)^2)^{\frac{1}{2}} + \sum_{j=1}^{N} \|x_j\|_2^2} \right\} \overset{\text{Jensen's ineq.}}{\leq}$$

$$\frac{1}{N} \sqrt{2d} G C_1 \cdot \left\{ \sqrt{C_2 \cdot \underset{X}{\mathbb{E}}(\sum_{1 \leq i < j \leq N} (x_i^T x_j)^2)^{\frac{1}{2}} + \underset{X}{\mathbb{E}} \sum_{j=1}^{N} \|x_j\|_2^2} \right\} \leq$$

$$\frac{1}{N} \sqrt{2d} G C_1 \cdot \left( C_2 \cdot (\sum_{1 \leq i < j \leq N} \underbrace{\underset{X}{\mathbb{E}}(x_i^T x_j)^2}_{\leq R^4})^{\frac{1}{2}} + N \underbrace{\underset{X}{\mathbb{E}}\|x_1\|_2^2}_{\leq R^2} \right)^{\frac{1}{2}} \leq \tag{66}$$

$$\frac{1}{N} \sqrt{2d} G C_1 \cdot \left( C_2 \cdot \sqrt{\frac{N(N-1)}{2}} R^2 + N R^2 \right)^{\frac{1}{2}} = O(\frac{1}{\sqrt{N}}).$$

It completes the proof. $\qquad\qquad\qquad\qquad\qquad\qquad\qquad\qquad\qquad\qquad\qquad\qquad\qquad\qquad\square$

# B  EXTENDED DISCUSSION ON THEORETICAL ASSUMPTIONS & LOWER BOUND ANALYSIS & PRACTICAL RELEVANCE

**Discussion on the $\beta$-strongly convexity assumption.** It seems we can not get rid of this assumption. Now we explain the roots of this below:

- If we allow in Theorem 4.6, that $\varphi^*$ is not $\beta$ - Lipschitz, we still may achieve $\mathcal{L}(\varphi_L^\beta) - \mathcal{L}(\varphi^*) < \varepsilon$, but for $\beta = C\varepsilon$, $C$ is some constant.
- This makes Theorem 4.1 not useful, because we restrict $L^2$ gap by $\frac{\varepsilon}{C\varepsilon} = \frac{1}{C}$, i.e., can not achieve arbitrary small gap.

Theoretically, $\beta$-strong convexity for $\varphi$ holds where the OT mapping between distributions is regular enough. By (Kakade et al., 2009) and our Lemma A.10 this means that $T_\varphi$ is Lipschitz, i.e., does not push close points far away from each other. We assume this holds true for reasonable (smooth, supported on the same set) underlying distributions.

**Lower bound analysis.** To start with, we need to explain how we position our paper compared to other works which conducts statistical analysis of OT (primarily, (Hütter & Rigollet, 2021), (Divol et al., 2025), (Gunsilius, 2022)). Overall, the general framework of these papers is as follows:

1. They place certain restrictions on source/target distributions and class of functions in which we are looking OT mapping;

2. Under these assumptions, they derive (minimax-optimal) rates;

3. Then they discuss or construct some family of approximators which to some extend satisfies the assumptions.

This framework provides a strong theoretical foundation, though one practical consideration is that the proposed approximators (which achieve the optimal rate) may differ from those typically adopted in real-world applications.

In our paper, we take a different approach. We do not place restrictive assumptions on the optimized class of functions (we pick MLPs, ICNNs), and analyze the minimax OT solvers, which seem to be *quite popular* in practice, see §2 for the list of the methods which are relevant to us.

We think that in our setting it is much harder to conduct study on convergence with more sharp rates. One reason is the complexity of saddle-point nature minimax solvers (the aforementioned papers study just minimization), the other is our weak restrictions on function classes. At the same time, we emphasize that our work seems to be the *first* which conducts statistical study of a popular and practically-appealing algorithm used to recover OT maps.

**Practical relevance.** In our paper, we study theoretical properties of certain Optimal Transport solvers. We insist that the ability to model OT maps rather than arbitrary push-forward maps is an important and practically inspired field of research. One of the most intriguing applications of OT is unpaired domain translation of images, see, e.g., (Korotin et al., 2023b; Shi et al., 2023). Here, thanks to the 'optimality' of the learned image-to-image mapping, the content of the transformed images remains similar to that of the originals. This could be crucial for image enhancement and super-resolution applications, for example. The other solid application of the OT is computational biology (Tong et al., 2020; Demetci et al., 2022; Bunne et al., 2023) (manipulating single-cell data). Here, the OT appears by nature of the problem; it turns out to be deeply connected with the development and proliferation of cell populations.

Importantly, all of these practical use cases have inspired the development of a variety of minimax OT solvers (close to our theoretical analysis), see the list in the end of §2.

## C  ADDITIONAL EXPERIMENTAL ILLUSTRATIONS

### C.1  ESTIMATION ERRORS OF BASELINE METHODS

In this section, we consider three non-neural baseline methods: barycenter translation estimator (*mean shift*), *constant* and *linear*. Suppose that $x$ and $y$ correspond to samples from distributions $p$ and $q$, respectively. Then for constant estimator the transport map is given by $T = \mathbb{E}_q(y) \equiv \mu_q$. For barycenter translation estimator, the transport map $T$ has the form $T(x) = x + (\mathbb{E}_q(y) - \mathbb{E}_p(x)) = x + \mu_q - \mu_p$. Finally, the linear estimator is given by:

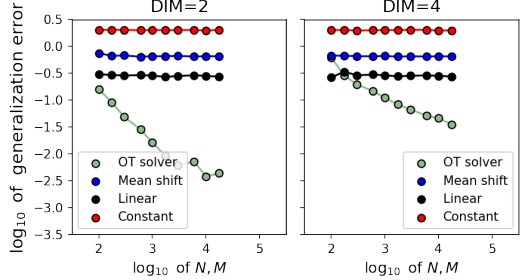

Figure 5: Estimation errors of the baseline solvers using a limited number of available empirical training samples $N, M$.

$$T(x) = \Sigma_p^{-\frac{1}{2}} \left( \Sigma_p^{\frac{1}{2}} \Sigma_q \Sigma_p^{\frac{1}{2}} \right) \Sigma_p^{-\frac{1}{2}} (x - \mu_p) + \mu_q.$$

We refer to (Korotin et al., 2021b) for the additional details on these baselines. We run the solvers in the experimental setup of §5.1 using the limited number of available training samples $N, M$ for *learning* the means and covariances of input and target distributions. The estimation errors are reported in Fig. 5.

The results show that the *linear* estimator achieves the strongest performance among all baselines. Still, the estimation errors of the OT solver are **significantly lower** than the errors of all baseline solvers. Besides, as expected, these non-neural estimators do not show noticeable dependence on the number of available training samples.

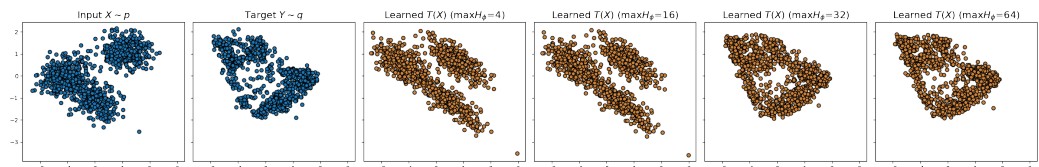

Figure 6: Convergence rates of the OT solver learned with the quadratic transport cost and a limited number of empirical training samples for distributions of dimensions $D = 8, 16, 32, 64, 128$.

Figure 7: Visualization of the solutions of OT solver under the experimental setup from §5.2 − DIM $= 2$, $\max H_T = 8$. The figure shows that the usage of shallow architectures for potential $H_\varphi$ leads to improper collapsing solutions.

## C.2 HIGH-DIMENSIONAL EXPERIMENTS

In this section, we report the estimation error of OT solver in dimensions $D = 8, 16, 32, 64, 128$. We follow the experimental setup described in §5.1 and visualize the results in Fig. 6. The **results** confirm our main conclusion that the estimation error of the *minimax* OT solver decreases with the increase of the number of empirical training samples $N, M$ w.r.t. $\log_{10}$-$\log_{10}$ scale.

Meanwhile, we plot the slope of convergence of the logarithm of the error w.r.t. the logarithms of $N, M$ and observe that it is not always less than $-0.5$. Besides, we do not observe any evident dependence between the dimension and the slope of convergence, e.g., the highest dimension $D = 128$ corresponds to the convergence with the steepest slope $< -0.58$. These observations could be explained by two main reasons. First, our Corollary 4.10 establishes the *upper bound* on the estimation error. Thus, while upper bound theoretically exhibits linear convergence with the slope equal to $-0.5$, it does not guarantee that the error itself exhibits the same rate of convergence. Second, as was noted in §5.1, the theoretical bounds might be corrupted by the optimization errors. These optimization errors might have different effects on the generalization error depending on the number of available training samples $N, M$; analysis of their roots and consequences represent an interesting future research direction.

## C.3 APPROXIMATION ERRORS FOR SHALLOW ARCHITECTURES

In our experiments illustrating the effects of the approximation error (§5.2), we discussed the roots of potential improper solutions and large approximation errors in the case of extremely shallow architectures for potential $\varphi_\theta$ and/or transport map $T_\omega$. For example, in our Fig. 4 (main text), one might notice that the approximation error is accumulating at specific points, see the values for $\max H_\varphi = 4, 16$ (D $= 2$, $\max H_T = 8$) or $\max H_T = 1, 2$ (D $= 4$, $\max H_\varphi = 4$).

To further clarify the results of the OT solver for the case of extremely shallow architectures, we visualize its solutions in Fig. 7. Specifically, we plot the results of OT solver trained in the experimental setup described in §5.2 for dimension $D = 2$ and the transport map $T_\omega$ with the maximum width of the hidden layer equal to $8$. According to the approximation error reported for this experiment in Fig. 4, the solutions are divided into two modes − the mode with higher error corresponds to extremely shallow potential architectures, s.t., $\max H_\varphi = 4, 16$. The mode with smaller error corresponds to experiments with wider architectures of potential $\varphi$, s.t., $\max H_\varphi = 32, 64$. Our Fig. 7 shows that the results of the OT solver with shallow architectures are indeed collapsing and represent bad approximations of the target samples which could be explained by the

low expressivity of the used nets. At the same time, wider nets provide fine approximations of the target samples and are slightly different to each other.

## D    EXPERIMENTAL DETAILS

**General remark.** We parametrize the transport map $T_\omega$ using MLP architecture with ReLU activations. For the potential $\varphi_\theta$, we use ICNN architecture with the quadratic skip connections. Note that in our theoretical investigations, we consider more traditional ICNNs with linear skip connections, see §4. However, we decided to consider another type of ICNN for potential parametrization in order to minimize the impact of the approximation error since these ICNNs were used to construct the benchmark pairs from (Korotin et al., 2021b). While we can not totally eliminate the approximation error coming from the parametrization of OT map $T_w$, we strive to decrease it by selecting rich NN architecture ($\approx 2$ mln of parameters). Importantly, ICNNs found their popularity in a number of specific (e.g., OT) and general-purpose generative models, see (Huang et al., 2021; Dugan et al., 2023; Kornilov et al., 2024; Amos, 2023; Mokrov et al., 2021).

**Other details.** Each experiment requires several hours of training on one A100 GPU. We use Adam optimizer (Kingma & Ba, 2014) with lr=1e-3. In §5.1, we use batch size equal to $\min(1024, N)$ for each dataset size $N = M$. In §5.2, we use batch size equal to $1024$. We run the solver for 10K steps, performing 10 inner iterations of updating weights of $T_\omega$ at each step. We calculate the error between the learned $\widehat{T}$ and true $T^*$ OT maps: $\|\widehat{T} - T^*\|_{L^2(\mathbb{P})}$ using $2^{12}$ test samples. We implement standard linear regression using `numpy.polyfit` package. For each experiment in §5 and Appendix C.2, we run it 3 times with different random seeds, log the errors and report their mean.

## E    GENERATIVE ADVERSARIAL NETWORKS: RELATION TO THE SEMI-DUAL OT SOLVERS & THEORETICAL ANALYSIS

The objective of the semi-dual minimax continuous OT solvers (6) resembles that of GANs (Goodfellow et al., 2014). This fact motivates us to discuss the relation of the OT solvers to GANs and review some of the existing theoretical results on GANs below.

### E.1    RELATION OF SEMI-DUAL MINIMAX CONTINUOUS OT SOLVERS TO GANS

While there are many papers on GANs (Pan et al., 2019) and a relatively large number of works studying the theoretical aspects of their objectives, they have a limited relevance to our study for several reasons. First, OT solvers and GANs pursue different goals. The main goal of OT solvers is to approximate true OT maps $T^*$, i.e., to learn a *particular* optimal mapping (generator) between the given source and target distributions; accordingly, our theoretical work focuses on the error of this approximation. Meanwhile, GANs are focused on learning an arbitrary mapping which pushforwards some noise to a target distribution; thus, existing theoretical results on GANs mostly investigate the error of approximating the ground-truth distribution $q$ by the generated $p_{gen}$. Second, OT solvers and GANs have an evident difference corresponding to the order of optimization over generator $G$ (map $T$) and discriminator $D$ (potential $\varphi$) in their objectives. In particular, for GANs, the optimization over the generator $G$ is done in the outer problem of their objective, while for OT solvers, the optimization over OT maps is conducted in the inner problem, i.e., the OT map $T^*$ corresponds to the solution of the inner problem. We emphasize the importance of this difference − for OT solvers, OT map $T^*$ corresponds to the solution of the inner problem, which makes the task of theoretical investigation of the error of $T^*$ approximation even more difficult.

Besides, we outline the theoretical and methodological differences between GANs and semi-dual OT.

**The origin of GANs training objective.** We have some noise $p_z$, a (parametric) generator model $G$ and target distribution $q$, and want $G_{\#}p_z = q$. To achieve this goal, we use a discrepancy, $\mathcal{D}(G_{\#}p_z, q)$, which compares generated and target distributions. Theoretically, the discrepancy could be anything, e.g., KL (JSD), MMD, Wasserstein distance, etc. The discrepancy ensures the generated and target distributions are close, but does not restrict $G$ to be of a particular form (i.e., any $G$ which maps $p_z$ to $q$ would result in zero discrepancy). Also, the order of practical optimization is also dictated by the explained scheme. Our aim is to solve $\min_G \mathcal{D}(G_{\#}p_z, q)$, i.e., the outer optimization

is w.r.t. $G$. The inner adversarial maximization, in turn, appears due to a particular variational form of the discrepancy $\mathcal{D}$.

**The origin of semi-dual OT** stems from objective eq. (3) in our paper. I.e., there is a known theoretical result that optimizing eq. (3) w.r.t. potentials $\varphi$ results in OT cost (therefore, the outer optimization of semi-dual OT is w.r.t. potential (or discriminator), not mapping (generator)). The other theoretical result states that *Optimal Transport mapping* between $p$ and $q$ is $\nabla\varphi^*(x)$, where $\varphi^*$ is the solution to eq. (3). Therefore, the final objective we study in our work, eq. (6), is almost the same as eq. (3), but with Fenchel conjugate appearing in (3) substituted with the auxiliary learned *argument* of Fenchel transform, which is known to be exactly $\nabla\overline{\varphi}$.

As we can see, the **origins** of semi-dual OT and GAN-based training drastically differ, making the distinction clearer.

### E.2 THEORETICAL ANALYSIS OF ADVERSARIAL GENERATIVE MODELS

Below we discuss two groups of theoretical results on GANs which are slightly relevant to us.

**Estimation error of distance between generated and true distributions**. The work of (Arora et al., 2017) introduces a notion of an $\mathcal{F}$-*distance* between the pair of measures w.r.t. a given class of discriminators $\mathcal{F}$:

$$d_{\mathcal{F}}(p, q) = \sup_{D \in \mathcal{F}} \mathbb{E}_{x \sim p} \psi(D(x)) + \mathbb{E}_{y \sim q} \psi(1 - D(y)) - 2\psi(\frac{1}{2}).$$

Here $\psi : [0, 1] \to \mathbb{R}$ is any monotone concave function. For certain classes of functions, these distances define an inner optimization problem for well-known GANs, e.g., Wasserstein, Vanilla, MMD, Sobolev, etc. Class $\mathcal{F}_{nn}$ corresponding to neural networks with a bound on a number of parameters yields a so-called *neural net (NN) distance*. The paper (Arora et al., 2017) establishes an upper bound on the estimation error of this distance, i.e., difference of NN distances $d_{\mathcal{F}_{nn}}(q, p_{gen})$ for empirical and continuous distributions. Subsequent work (Ji & Liang, 2018) extends this analysis by providing a tighter upper bound and lower bound for this difference. In some sense, these works analyze the error in GANs inner optimization problem coming from the usage of empirical measures $\widehat{p}, \widehat{q}$ instead of the real ones. Still, these works ignore the errors rooted in outer optimization problem of GANs.

**Estimation error of GANs.** Other class of papers takes into account errors in GANs' *outer* optimization problem. They commonly consider the solutions of GANs' empirical optimization problem: $\widehat{w}^* = \arg\min_{w \in \Omega} d_{\mathcal{F}_{nn}}(\widehat{q}, \widehat{p}_{gen(w)})$ where $\widehat{p}_{gen(w)}$ denotes the distribution generated from empirical samples of distribution $\widehat{p}$ by a generator $G_w$ (with parameters $w$ from the class $\Omega$). Then the papers derive upper bounds for the estimation error defined as

$$d_{\mathcal{F}}(q, p_{gen(\widehat{w}^*)}) - \inf_{w \in \Omega} d_{\mathcal{F}}(q, p_{gen(w)})$$

for different variants of function classes $\mathcal{F}$. For example, (Ji et al., 2021) derives an upper and minimax lower bound for this error and $\mathcal{F}_{nn}$ class; (Zhang et al., 2018) considers the same class but a different type of error using $w^* = \arg\min_{w \in \Omega} d_{\mathcal{F}}(\widehat{q}, p_{gen(w)})$ where $p_{gen(w)}$ is the distribution generated by $G_w$ from $p$; (Liang, 2017) and (Singh et al., 2018) study the same type of error but consider the discriminator class in Sobolev space; (Uppal et al., 2019) extends these result by modeling the discriminator and generator as Besov function classes. In general, these bounds are constructed based on Rademacher complexities of function spaces and are utilized to construct the bounds on the distance between $p_{gen}$ and $q$.

For completeness, we mention several other works which investigate theoretical properties of specific GANs, e.g., Vanilla (Biau et al., 2020), bidirectional (Liu et al., 2021), Wasserstein (Biau et al., 2021) and CycleGAN (Chakrabarty & Das, 2022), (Sun et al., 2024), (Moriakov et al., 2020).

## F  LLM USAGE

LLMs have been used only for polishing of the text purposes, i.e., for checking grammar and spelling, and for assistance with sentences rephrasing.

