# OpenReview forum: "A Statistical Learning Perspective on Semi-dual Adversarial Neural Optimal Transport Solvers"
_ICLR.cc/2026/Conference — ICLR 2026 Poster_

### Official Review · Reviewer_1LsA · 2025-10-31

**Soundness:** 2
**Presentation:** 2
**Contribution:** 2
**Rating:** 4
**Confidence:** 3

**Summary:**

The authors study a particular semi-dual formulation of the quadratic Optimal Transport (OT) problem and derive statistical guarantee bounds for recovering the true OT map when using a solver that optimizes this formulation. More precisely, they adopt a generative modeling framework and assume that the input and target distributions are absolutely continuous probability measures given through samples. They consider the case where the solver relies on two neural networks optimized using the semi-dual objective, which is approximated via Monte Carlo estimation based on the empirical distributions of the measures. The authors provide an upper bound on the L2 distance between the true OT map and the best transport map that could be obtained by the considered solver. To achieve this, they first decompose this gap, referred to as the generalization error in the paper, into two terms: the estimation error, which accounts for the discretization of the semi-dual formulation, and the approximation error, which arises because the optimization is performed over a restricted class of neural networks rather than the entire space of functions. They bound the first term using the Rademacher complexity of the function class and show that the second term can be made arbitrarily small by choosing an appropriate class of neural networks.

**Strengths:**

The theoretical results presented in this paper are interesting: the authors derive an upper bound on the number of samples required to approximate the true OT map using the proposed solver. Section 4, which focuses on deriving this upper bound, is well organized and clearly written, which permits to easily follow the main steps of the derivation.

**Weaknesses:**

- I am a bit perplexed by the experimental setup. In Section 5.1, the authors assume that, because they use the same neural network architecture as the one corresponding to the ground-truth OT map, there is no gap in the approximation error, and thus the observed generalization error corresponds solely to the estimation error. This appears to be a strong assumption, as it implies that the optimization process finds the global minimum of the loss defined in Equation (8) and does not get trapped in local minima when training the ICNN.

- Regarding the writing, Sections 2 and 3 could be improved. The theoretical background on OT with general cost functions is not necessary to understand the paper’s contributions. The authors should focus exclusively on the quadratic cost in Section 2. Similarly, the paragraph “Continuous OT solvers for quadratic cost” does not seem essential for understanding the main contributions.

- Finally, the writing should be polished and carefully revised for clarity and precision (see, for example, lines 352–353, 405–406, and 410–411).

**Questions:**

- Does the class of neural networks described in Proposition 4.4 represent any $\beta$-strongly convex function that is totally bounded with respect to the Lipschitz norm?

- What does assuming that the optimal potential $\phi^*$ is $\beta$-strongly convex imply about the considered distributions $p$ and $q$ ?

- In Section 5.1, you assume that, because you use the same neural network architecture as the one corresponding to the ground-truth OT map, there is no gap in the approximation error, and that the observed generalization error is, in fact, the estimation error. This seems to be a strong assumption, as it implies that the optimization process reaches the global minimum of the loss defined in Equation (8) and does not get trapped in local minima when optimizing the ICNN (a similar comment applies to Section 5.2). Could you comment on this? Do you observe that, for different initializations of the ICNN, the optimization consistently recovers approximately the same transport map? Or, did you try initializing the neural network with several random seeds and retaining the best result?

- Your experiment in Section 5.1 seems ideal, as it suggests that the bound you derive is tight in dimension 4—is that correct? I am not a specialist in statistical bounds, but is it common to verify experimentally that such bounds are tight as you did? Could you elaborate on this?

- During the experiments, did you constrain the ICNN as described in Proposition 4.4? Do you observe that imposing these constraints helps the optimization in practice?

- Finally, there is a small typo: it should be $\varepsilon / \beta$ in Equation (20), although this does not affect the result.

**Details Of Ethics Concerns:**

no concerns

---

> ### Author Response · Authors · 2025-11-22
> **Answer to reviewer 1LsA (1/3)**
>
> Thank you for your thorough feedback. Please find the answers to your questions below.
>
> **[(W1) I am a bit perplexed by the experimental setup. In Section 5.1, the authors assume that, because they use the same neural net architecture as the one corresponding to the ground-truth OT map, there is no gap in the approximation error, and thus the observed generalization error corresponds solely to the estimation error. It appears to be a strong assumption, as it implies that the optimization process finds the global minimum of the loss defined in Equation (8) and does not get trapped in local minima when training the ICNN.<...>(Q3) Could you comment on this? Do you observe that, for different initializations of the ICNN, the optimization consistently recovers approximately the same transport map? Or, did you try initializing the neural net with several random seeds and retaining the best result?**
>
> We thank the reviewer for the question. We want to clarify that in our section 5.1 we do not claim that the empirical generalization error is strictly equal to the estimation error. We agree that such a claim will be too strong, at least because of the possible optimization errors which we mention in sections 5.1/5.2 of our paper. Still, in section 5.1, we do our best to minimize the gap in the approximation error and capture the overall trend of generalization error irrespective (as much as possible) to optimization issues. For this purpose, for each of the experiments in section 5.1 (and 5.2 also), we trained the networks with 3 random seeds, log the errors and report their means in Fig. 3 (and Fig. 4, respectively). Overall, the trends and transport maps over these 3 different initializations of the networks appeared to be approximately the same. For example, in the case of DIM=4, standard deviation of the errors before application of $\log$-transformation constitutes $\approx 10$% of the mean. In our new experiments in high dimensions (see our new Appendix C.2), these variability becomes even lower, e.g., for DIM$=128$, standard deviation of the error constitutes $\approx 4$% of the mean.
>
> *For clarity, we include the information about random seeds into the revised version of our paper (Appendix D).*
>
>
> **(W2) Regarding the writing, Sections 2 and 3 could be improved. The theoretical background on OT with general cost functions is not necessary to understand the paper’s contributions. The authors should focus exclusively on the quadratic cost in Section 2. Similarly, the paragraph “Continuous OT solvers for quadratic cost” does not seem essential for understanding the main contributions.**
>
> Following the reviewer's suggestion, we considerably shortened Section 2 (see the revised manuscript):
>
> - Paragraph "Semi-dual OT problem" (contains some non-vital background of the OT with general cost function) and Remark before this paragraph: completely removed;
>
> - Paragraph "Semi-dual OT problem with quadratic cost": modified to include necessary details from the removed parts.
>
> Additionally, we shortened the paragraph "Continuous OT solvers for quadratic cost". We humbly think that removing this paragraph completely will negatively affect our work, since there we
> **(a)** introduce some concepts used in the subsequent text (e.g., ICNNs); **(b)** introduce a basic strategy how people solve OT with quadratic cost function, which is important to properly *position* our developed framework among the alternatives. In particular, compared to our considered min-max OT, this strategy (dual + potential-only) was extensively analyzed with statistical framework, see section 3.
>
> **(Q1) Does the class of neural networks described in Proposition 4.4 represent any beta-strongly convex function that is totally bounded with respect to the Lipschitz norm?**
>
> We would like to clarify the reviewer's question. We assume that it concerns the "richness" of class $\mathcal{F}$ appearing in Proposition 4.4. To avoid possible misunderstanding, we humbly note the property to be "totally bounded with respect to the Lipschitz norm" (meaning the existence for each $\epsilon > 0$ an $\epsilon$-net) is applicable only to a class of functions, not to a particular function itself. Based on this, we interpret the question as "Does the class of neural networks described in Proposition 4.4 represent any beta-strongly convex function?"
>
> Remarkably, the answer to this question is **yes**, and it follows exactly from our theoretical derivations. In particular - from the proof of Theorem 4.6. Lines 1416-1418 of our manuscript (revised version) state that
> $\Vert \varphi^* - \varphi\_L^{\beta}\Vert\_{\infty} \leq \frac{\varepsilon}{2}$, where $\varphi^\*$ is an arbitrary $\beta$-strongly convex function, and $\varphi\_L^{\beta}$ comes exactly from class $\mathcal{F}$, described in Prop. 4.4. Importantly, the choice of class $\mathcal{F}$ **does not** depend on $\varphi^*$, but only on the strong convexity coefficient $\beta$ and desired precision $\varepsilon$.

---

> ### Author Response · Authors · 2025-11-22
> **Answer to reviewer 1LsA (2/3)**
>
> **(W3) Finally, the writing should be polished and carefully revised for clarity and precision (see, for example, lines 352–353, 405–406, and 410–411).**
>
> Thank you for your attentive review. Could you please elaborate on what exactly seems unclear in those lines? We will be happy to revise the text accordingly if you could provide us the details.
>
> **(Q2) What does assuming that the optimal potential  phi is beta-strongly convex imply about the considered distributions  p and  q?**
>
> Thank you for this question - in the revised version of our manuscript we add a Remark (4.8) regarding $\beta$-strong convexity assumption on the optimal ground truth potential $\varphi^*$: under certain assumptions on $p$ and $q$ $\beta$-strong convexity does hold.
>
> In particular, following [1, Theorem ([C3])], it holds that
>
> - if data supports $\mathcal{X}$, $\mathcal{Y}$ are convex;
>
> - probability distribution densities, $p$ and $q$ are strictly positive and bounded on $\mathcal{X}$ and $\mathcal{Y}$, respectively;
>
> - $p$ and $q$ are $C^{\alpha}$ for some $\alpha > 0$, i.e., Holder continuous functions: $\exists \alpha > 0$, $C > 0$, $\forall x_1, x_2 \in \mathcal{X}; \forall y_1, y_2 \in \mathcal{Y}$: $\vert p(x_1) - p(x_2) \vert \leq C \Vert x_1 - x_2 \Vert^{\alpha}$; $\vert q(y_1) - q(y_2) \vert \leq C \Vert y_1 - y_2 \Vert^{\alpha},$
>
> then the optimal potential $\varphi^*$ is indeed strongly convex, and we can verifiably apply our developed theoretical framework.
>
> Noteworthy, the Holder continuity does not seem to be restrictive assumption: it holds for majority of "tabular" distributions, mixture of distributions (e.g., Gaussians), which are known to be universal approximators.
>
> [1] Caffarelli, L. A. (1996). Boundary regularity of maps with convex potentials--II. Annals of mathematics, 144(3), 453-496.
>
> **(Q4) Your experiment in Section 5.1 seems ideal, as it suggests that the bound you derive is tight in dimension 4—is that correct? I am not a specialist in statistical bounds, but is it common to verify experimentally that such bounds are tight as you did? Could you elaborate on this?**
>
> To start with, we would like to clarify that our experimental analysis of errors is not a common technique especially in the context of practically working algorithms which usually depend on tons of hyper-parameters, complex optimization, etc. Many of the classic related works exploring error rates of computational OT algorithms with neural parametrization (Gonzalez-Sanz, 2022; Divol, 2025; Gunsilius, 2022) provide theoretical results without experimental illustrations. Still, we found rather similar to ours experimental illustrations in the related papers on non-minimax OT solvers, see, e.g., Fig. 2 in (Hütter \& Rigollet, 2021), Table 3 in (Ding, 2024). Meanwhile, we received considerable interest from the ML community and from other reviewers regarding the inclusion of such experiments, see, e.g., the questions of Reviewers BuZV, aRUC, which motivated us to perform the experimental illustrations.
>
> Regarding the experiment in section 5.1, DIM=4 - it was not aimed at showing that our theoretical bound for the estimation error is tight. Actually, the goal of all our experimental illustrations consists in confirming our main conclusion that the estimation error of the *minimax* OT solver decreases (in $\log_{10}$-$\log_{10}$ scale) with the increase of the number of empirical training samples $N,M$. In the revised version of our paper, we have added additional experiments in higher dimensions (from DIM=8 to DIM=128), see our new Appendix C.2, which also justify this conclusion.
>
> (Gonzalez-Sanz, 2022) Alberto Gonzalez-Sanz, Lucas De Lara, Louis Bethune, and Jean-Michel Loubes. Gan estimation of
> lipschitz optimal transport maps. arXiv preprint arXiv:2202.07965, 2022.
>
> (Hütter \& Rigollet, 2021) Hütter, J. C., \& Rigollet, P. (2021). Minimax estimation of smooth optimal transport maps.
>
> (Gunsilius, 2022) Gunsilius, F. F. (2022). On the convergence rate of potentials of Brenier maps. Econometric Theory, 38(2), 381-417.
>
> (Ding, 2024) Ding, Y., Li, R., \& Xue, L. (2024). Statistical convergence rates of optimal transport map estimation between general distributions. arXiv preprint arXiv:2412.08064.
>
> (Divol, 2025) Divol, V., Niles-Weed, J., \& Pooladian, A. A. (2025). Optimal transport map estimation in general function spaces. The Annals of Statistics, 53(3), 963-988.

---

> ### Author Response · Authors · 2025-11-22
> **Answer to reviewer 1LsA (3/3)**
>
> **(Q5) During the experiments, did you constrain the ICNN as described in Proposition 4.4? Do you observe that imposing these constraints helps the optimization in practice?**
>
> In our experiments, the $\beta$-strong convexity of our ICNN architecture is enforced by design (exactly by $\beta \frac{\Vert x \Vert^2}{x}$ added to the output of the neural network with some small pre-defined $\beta$ ($\beta = 10^{-4}$).
>
> Regarding the compact weights constraint - the situation is more tricky. We did not specifically impose the boundedness of weight matrices $\Vert a_i \Vert_2 \leq A$ for some specific pre-defined $A$. The reason is that, in any case, we cannot ideally reproduce the theoretical setup in the experiments. Even if the class of potentials $\mathcal{F}$ satisfy Proposition 4.4, it is not trivial to derive the class of maps $\mathcal{T} = \mathcal{T}(\varepsilon, \mathcal{F})$ which yields controllable inner approximation error as per Theorem 4.3.
>
> Following common sense, in our experiments we pick reasonable architectures both for $\mathcal{F}$ and $\mathcal{T}$, please see \S 5 for the details. As we did not observe any ''weight explosions'', the ''compactness'' is expected to be satisfied to some extent.
>
> We kindly ask the reviewer whether our answer addresses their question.
>
> **(Q6) Typo in Equation (20) - it should be $\frac{\varepsilon}{\beta}$.**
>
> We thank the reviewer for the thorough attention to details. Please note that $\varepsilon$ in our Equation (20) could be chosen in such a way that the constant $\beta$ is hidden inside it. In the revised version of our paper, we have updated the proof of corresponding Theorem 4.9 (revised manuscript) which now fully supports this explanation.
>
>  **Concluding remarks**
>
> Thank you for your thoughtful feedback. We would appreciate it if you could let us know whether our clarifications address your concerns. If there are any remaining issues, we would be happy to discuss them further. Otherwise, we kindly ask you to consider updating your score accordingly.

---

> > ### Comment · Reviewer_1LsA · 2025-11-26
> >
> > Dear authors,
> >
> > Thank you for your answers.
> >
> > You addressed my concerns/questions for W1, Q1, Q5, and Q6. I think that the rewriting of Section 2 is improved now, and there are no « superfluous » elements. Regarding W3, I was simply mentioning that the experimental section should be reread. For example, the following sentence does not contain any verb: “For clarity of illustration, we study these rates for the examples where the generalization error in the proximity of either the estimation error (§5.1) or the approximation error (§5.2).” This is not a major issue, but I noticed similar errors elsewhere in the first version of the paper. I also saw that you added clarifications concerning the implications of assuming that (p) and (q) are (\beta)-strongly convex, and I think this is a good point.
> >
> > Concerning Q4, I read the corresponding paragraph in Appendix C.2. If I understand correctly, Corollary 4.10 ensures that the slope you should obtain on the graphs is less than −0.5. However, in the experiments shown in Figure 6, you obtain slopes greater than −0.5. You say that the reason for that is that it is difficult to only extract the estimation error. In particular, you give two possible sources of errors, the first one is the optimization error. Regarding the second reason, what exactly do you mean by: “First, our Corollary 4.10 establishes the upper bound on the estimation error. Thus, while upper bound theoretically exhibits linear convergence with the slope equal to −0.5, it does not guarantee that the error itself exhibits the same rate of convergence.” Do you mean that the error in your plot also includes the approximation error, and therefore it is expected that the observed slopes are greater than −0.5? In that case, why would the experiments be meaningful for observing a linear decrease (in log-log scale) of the estimation error?

---

> > > ### Author Response · Authors · 2025-11-28
> > > **Additional response to Reviewer 1LsA**
> > >
> > > Dear Reviewer,
> > >
> > > Thank you for the involvement in the discussion. Please find our additional answers below.
> > >
> > > **Regarding W3, I was simply mentioning that the experimental section should be reread. For example, the following sentence does not contain any verb: “For clarity of illustration, we study these rates for the examples where the generalization error in the proximity of either the estimation error (section 5.1) or the approximation error (section 5.2).” This is not a major issue, but I noticed similar errors elsewhere in the first version of the paper.**
> > >
> > > We thank the reviewer for providing the additional details on the requested text clarifications. We have revised the text of our paper in the noted lines 352-353, 405-406, 410-411 and the experimental section. All *new* changes are highlighted in **orange** colour.
> > >
> > > **Concerning Q4, I read the corresponding paragraph in Appendix C.2. If I understand correctly, Corollary 4.10 ensures that the slope you should obtain on the graphs is less than -0.5. However, in the experiments shown in Figure 6, you obtain slopes greater than -0.5. You say that the reason for that is that it is difficult to only extract the estimation error. In particular, you give two possible sources of errors, the first one is the optimization error. Regarding the second reason, what exactly do you mean by: "First, our Corollary 4.10 establishes the upper bound on the estimation error. Thus, while upper bound theoretically exhibits linear convergence with the slope equal to -0.5, it does not guarantee that the error itself exhibits the same rate of convergence."**
> > > - **Do you mean that the error in your plot also includes the approximation error, and therefore it is expected that the observed slopes are greater than -0.5?**
> > > - **In that case, why would the experiments be meaningful for observing a linear decrease (in log-log scale) of the estimation error?**
> > >
> > > We would like to provide additional clarifications on the results of our new high-dimensional experiments  in Appendix C.2. The results of these experiments indeed show that the experimental error (visualized on Fig. 3 and Fig. 6) might exhibit the convergence with the rate lower than $-0.5$. Strictly speaking, this experimental error does not correspond solely to the estimation error, but rather to the composition of the estimation error, approximation error and the uncontrollable optimization one. While in our experiments in section 5.1 and Appendix C.2 we did our best to minimize the approximation and optimization errors, e.g., by choosing the  relevant architectures of the networks (see lines 414-415), these errors still might influence the revealed trend of the error. Thus, we attribute the differences between the obtained theoretical and experimental results mainly to these additional errors. Note that similar difference between the theoretical and experimental convergence rates was noted in the paper of (Hütter \& Rigollet, 2021) which includes the experiments similar to ours (their Fig. 2). The authors of this paper also explain this difference by the additional errors, i.e., optimization error, corrupting the results.
> > >
> > > Still, the main goal of our paper is to show the **statistical consistency** of the neural minimax OT solver. Our corollary 4.10 justifies this fact from a theoretical point of view. Meanwhile, our experiments in section 5.1 and Appendix C.2 also support this conclusion by showing that the generalization error **decreases** with the increase of the number of empirical samples $N$ and $M$. Interestingly, we also empirically revealed that the experimental error exhibits linear convergence w.r.t. $\log_{10}$-$\log_{10}$ scale.
> > >
> > > (Hütter \& Rigollet, 2021) Hütter, J. C., \& Rigollet, P. (2021). Minimax estimation of smooth optimal transport maps.

---

### Official Review · Reviewer_aRUC · 2025-11-03

**Soundness:** 3
**Presentation:** 3
**Contribution:** 3
**Rating:** 6
**Confidence:** 3

**Summary:**

The paper provides a theoretical characterization of the generalization error for optimal transport mappings parameterized by input-convex neural networks, focusing on a class of continuous Wasserstein-2 distance solvers. The analysis decomposes the overall error into two components, estimation error and approximation error, each of which is independently derived and subsequently combined to yield the final generalization bound. The theoretical findings are supported by experimental validation on a simple, low-dimensional synthetic dataset, which confirm the consistency of the results with the proposed theoretical framework.

**Strengths:**

The paper makes an original theoretical contribution by deriving generalization error bounds for quadratic optimal transport (OT) solvers parameterized by neural networks, particularly within the class of input-convex models. This work provides valuable insights into the learnability and error behavior of neural OT mappings and offers practical guidance for their use in learning-based transport problems. The theoretical framework is rigorous, decomposing the overall generalization error into estimation and approximation components with clear mathematical justification.

The paper is clearly written, logically structured, and easy to follow, even when presenting technically complex material. Although the experiments are limited to simple synthetic datasets, they effectively validate the theoretical results. The significance of this work lies in establishing foundational learnability guarantees for neural OT solvers, which bridge the gap between theoretical understanding and practical implementation in modern OT-based learning frameworks.

**Weaknesses:**

A key limitation of the paper is its exclusive reliance on low-dimensional synthetic datasets for empirical validation. While the theoretical analysis is sound, the absence of experiments on higher-dimensional or real-world datasets limits the assessment of the framework’s practical relevance and robustness. Extending the experiments to more complex domains would provide stronger empirical support for the theoretical claims and demonstrate the scalability of the proposed bounds.

Additionally, the paper assumes \beta-strong convexity, a condition that is often difficult to guarantee in neural network parameterizations. While the authors acknowledge this assumption as restrictive, a deeper discussion of its implications and potential violations would be valuable.

Some missing references, e.g. [proof ref.] in multiple theorems and propositions.

**Questions:**

In Figure 3, the slope appears to increase as the dimensionality rises. Could the authors provide further insights into the underlying reasons for this behavior? Additionally, as the dimensionality continues to grow, do the authors expect the slope to remain below or equal to 0.5?

---

> ### Author Response · Authors · 2025-11-22
> **Answer to reviewer aRUC**
>
> Thank you for your thorough feedback. Please find the answers to your questions below.
>
> **[(W1) A key limitation of the paper is its exclusive reliance on low-dimensional synthetic datasets for empirical validation. While the theoretical analysis is sound, the absence of experiments on higher-dimensional or real-world datasets limits the assessment of the framework’s practical relevance and robustness. Extending the experiments to more complex domains would provide stronger empirical support for the theoretical claims and demonstrate the scalability of the proposed bounds.<...> (Q1) In Figure 3, the slope appears to increase as the dimensionality rises. Could the authors provide further insights into the underlying reasons for this behavior? Additionally, as the dimensionality continues to grow, do the authors expect the slope to remain below or equal to 0.5?]**
>
> To mitigate the concerns of the reviewer, we study the estimation error of OT solver in higher dimensions from $D=8$ to $D=128$. In these experiments, we follow the experimental setup established in our section 5.1. In the revised version of our paper, *we have included the results of these higher-dimensional experiments in the new Appendix C.2.*
>
> The **results** confirm our main conclusion that the estimation error of the *minimax* OT solver decreases (in $\log_{10}$-$\log_{10}$ scale) with the increase of the number of empirical training samples $N,M$.  Meanwhile, we plot the slope of convergence of the logarithm of the error w.r.t. the logarithms of $N,M$ and observe that it is not always less than $-0.5$. Besides, we do not observe any evident dependence between the dimension and the slope of convergence, e.g., the highest dimension $D=128$ corresponds to the convergence with the steepest slope $<-0.58$. These observations could be explained by two main reasons. First, our Corollary 4.10 (revised manuscript) establishes the *upper bound* on the estimation error. Thus, while upper bound theoretically exhibits linear convergence with the slope equal to $-0.5$, it does not guarantee that the error itself exhibits the same rate of convergence. Second, as was noted in section 5.1, the theoretical bounds might be corrupted by the optimization errors. These optimization errors might have different effects on the generalization error depending on the number of available training samples $N,M$; analysis of their roots and consequences represent an interesting future research direction.
>
> **[(W2) Additionally, the paper assumes $\beta$-strong convexity, a condition that is often difficult to guarantee in neural network parameterizations. While the authors acknowledge this assumption as restrictive, a deeper discussion of its implications and potential violations would be valuable.**
>
> We thank the reviewer for this valuable comment. In the revised version of our manuscript we add a Remark (4.8) regarding $\beta$-strong convexity assumption on the optimal ground truth potential $\varphi^*$. We show, that this assumption is not very restrictive, and holds true under some reasonable assumptions on the source and target distributions $p$ and $q$.
>
> In particular, following [1, Theorem ([C3])], it holds that
>
> - if data supports $\mathcal{X}$, $\mathcal{Y}$ are convex;
>
> - probability distribution densities, $p$ and $q$ are strictly positive and bounded on $\mathcal{X}$ and $\mathcal{Y}$, respectively;
>
> - $p$ and $q$ are $C^{\alpha}$ for some $\alpha > 0$, i.e., Holder continuous functions: $\exists \alpha > 0$, $C > 0$, $\forall x_1, x_2 \in \mathcal{X}; \forall y_1, y_2 \in \mathcal{Y}$: $\vert p(x_1) - p(x_2) \vert \leq C \Vert x_1 - x_2 \Vert^{\alpha}$; $\vert q(y_1) - q(y_2) \vert \leq C \Vert y_1 - y_2 \Vert^{\alpha}$,
>
> then the optimal potential $\varphi^*$ is indeed strongly convex, and we can verifiably apply our developed theoretical framework.
>
> Noteworthy, the Holder continuity does not seem to be restrictive assumption: it holds for majority of "tabular" distributions, mixture of distributions (e.g., Gaussians), which are known to be universal approximators.
>
> [1] Caffarelli, L. A. (1996). Boundary regularity of maps with convex potentials--II. Annals of mathematics, 144(3), 453-496.
>
> **[(W3) Some missing references, e.g. [proof ref.] in multiple theorems and propositions.**
>
> We want to clarify that [proof ref.] links (they are ''clickable'') in the formulations of the theorems lead to the specific sections in Appendix with the corresponding proofs. However, we agree with the reviewer that it might lead to ambiguities and have removed these links in the revised version of our paper.
>
> **Concluding remarks**
>
> Thank you for your thoughtful feedback. We would appreciate it if you could let us know whether our clarifications address your concerns. If there are any remaining issues, we would be happy to discuss them further. Otherwise, we kindly ask you to consider updating your score accordingly.

---

### Official Review · Reviewer_BuZV · 2025-11-04

**Soundness:** 3
**Presentation:** 4
**Contribution:** 2
**Rating:** 6
**Confidence:** 4

**Summary:**

The paper studies the approximation capabilities of a certain class of optimal transport problems for quadratic cost.  In particular, it focuses on the estimation of the *generalization* error: this stem from several factors, the approximation error (due to the dual potential being parametrized with a neural network) and the estimation error (stemming from finite sample).

This family of problems (and their solutions) is well studied, but the case of semi-dual solvers is less studied. Here, the semi-dual formulation allows to parametrize a potential using a convex class of functions, the ICNN.

The experimental part of the paper study a particular implementation of the approach using Input Convex Neural Network (ICNN).

I did not check the proofs in appendix

**Strengths:**

The paper is well written, very clear, and accessible to people with familiarity in optimal transportation, learning theory and statistical consistency. Overall, the strategy to tackle the problem is well explained.

The related work is well done, especially on semi-dual formulations, which are a getting popular, making the contribution timely.

The methodology is standard, but the result is new. Only a subset of results actually depend on the approximation property of certain neural networks classes. Others are a classical decomposition of the error between generalization and approximation.

We can appreciate the diversity of tools and techniques used.

**Weaknesses:**

### Missing related work

The recent work of Nietert and Goldfeld is relevant and could be added to the related work.

Nietert, S. and Goldfeld, Z., *Estimation of Stochastic Optimal Transport Maps*. In The Thirty-ninth Annual Conference on Neural Information Processing Systems, 2025.

### Somewhat unconvincing experiments

My enthusiasm for the paper is somewhat tempered by the numerical results.

**In Figure 3**, the logarithmic scale hides many phenomena. For example, I'd be curious to see what the experimental error looks like with the naive barycenter translation estimator: $T(x)=x+(\mathbb{E}_Q[x]-\mathbb{E}_P[x])$.

I have similar concerns with Figure 4 (see questions).

Overall, since some experiments do not depend on neural networks, and since ICNN are notably hard to train, it could be good to plot results of other (possibly non-neural) parametrizations.

**Questions:**

### Approximation power as function of depth

**In Figure 4, Dim 2**, I spot a counter-intuitive behaviors. I see that the approximation error does not diminish with $\max H_{\phi}$. Do you know why? This somewhat contradicts Theorem 4.3.

Can you clarify what is the **x-axis**: why does it cover the 0-1 range?

Finally, we sometimes see the error accumulating on special values (e.g Dim 2 maximum width, or Dim 4 small width). Can you double check that the network is not degenerating toward a trivial solution?

---

> ### Author Response · Authors · 2025-11-22
> **Answer to reviewer BuZV (1/2)**
>
> Thank you for your thorough feedback. Please find the answers to your questions below.
>
> **[(W1) The recent work of Nietert and Goldfeld is relevant and could be added to the related work.]**
>
> We thank the reviewer for pointing to the relevant paper. We  included it in the revised version of our paper.
>
> **[(W2) ... since some experiments do not depend on neural networks, and since ICNN are notably hard to train, it could be good to plot results of other (possibly non-neural) parametrizations. <...> (W3) I'd be curious to see what the experimental error looks like with the naive barycenter translation estimator]**
>
> We would like to clarify that all experiments conducted in our paper (sec. 5.1, 5.2) employ neural parametrization for the map - $T_{\omega}$, and potential - $\varphi_{\theta}$. Actually, the entire focus of our paper is on the derivation of the theoretical bounds for the errors of minimax OT solver which uses *neural parametrization*. Still, to address the question of the reviewer, we perform comparison with several non-neural baselines (estimators): (1) the 'barycenter translation estimator' suggested by the reviewer; (2) several other non-neural baselines established in (Korotin, 2021). *In the revised version of our paper, we have included the results of these experiments in our new Appendix C.1.*
>
> Overall, the results show that the *linear* estimator achieves the strongest performance among all baselines. Still, the estimation errors of the OT solver are **significantly lower** than the errors of all baseline solvers. Besides, as expected, these non-neural estimators do not show noticeable dependence on the number of available training samples.
>
> (Korotin, 2021) Korotin, A., Li, L., Genevay, A., Solomon, J. M., Filippov, A., \& Burnaev, E. (2021). Do neural optimal transport solvers work? a continuous wasserstein-2 benchmark. Advances in neural information processing systems, 34, 14593-14605.
>
> **[(Q1) "Approximation power as function of depth.
> In Figure 4, Dim 2, I spot a counter-intuitive behaviors. I see that the approximation error does not diminish with $\max H_{\varphi}$. Do you know why? This somewhat contradicts Theorem 4.3."]**
>
> We would like to clarify that our Theorem 4.3 does not state that the approximation error monotonically decreases to zero (with the growth of architectures). Still, it guarantees that we always could choose sufficiently expressive architectures for the transport map $T_{\omega}$ and potential $H_{\varphi}$ to make the approximation error as small as needed. Our experiment in section 5.2, Figure 4, is aimed to demonstrate this property. That is, we (a) show that the approximation error might severely affect the performance of the solver when the chosen architectures are too shallow; (b) confirm that we could make this error rather small by choosing the sufficiently wide architectures.
>
> **[(Q2) Can you clarify what is the x-axis: why does it cover the 0-1 range?]**
>
> Thank you for the question. We accidentally used incorrect tick marks for x-axis in our Fig. 4. Actually, this axis corresponds to the maximum width of the hidden layer of the transport map $T_{\omega}$ architecture which covers the range between 1 and 8 (as described in the experimental setup of section 5.2). *We corrected the tick marks in the revised version of our paper.*

---

> ### Author Response · Authors · 2025-11-22
> **Answer to reviewer BuZV (2/2)**
>
> **[(Q3) Finally, we sometimes see the error accumulating on special values (e.g Dim 2 maximum width, or Dim 4 small width). Can you double check that the network is not degenerating toward a trivial solution?]**
>
> Thank you for your suggestion. To clarify the performance of OT solver with shallow architectures, *we have plotted its results and analyzed them in new Appendix C.3 in the revised version of our paper.* Below we explain the error accumulation in our Fig. 2 on the example of DIM$=4, H_T=8$.
>
> According to the approximation error reported in our Fig. 4, the solutions are divided into two modes. Here the mode with higher error corresponds to extremely shallow potential architectures, s.t., $\max H_{\varphi}=4,16$. The mode with smaller error corresponds to experiments with wider architectures of potential $\varphi$, s.t., $\max H_{\varphi}=32,64$. Our newly added Fig. 7 in Appendix C.3 reveals that the results of the OT solver with shallow architectures ($\max H_{\varphi}=4,16$) indeed seem to be collapsing to improper solutions, i.e., representing bad approximations of the target samples. Note that this is an expected behavior since the nets
> with too shallow architectures have very limited expressivity which might lead to improper solutions and large approximation errors. At the same time, **wider nets provide fine approximations** of the target samples and are slightly different to each other, i.e., visual exploration shows that there is no collapse in this case.
>
> **Concluding remarks**
>
> Thank you for your thoughtful feedback. We would appreciate it if you could let us know whether our clarifications address your concerns. If there are any remaining issues, we would be happy to discuss them further. Otherwise, we kindly ask you to consider updating your score accordingly.

---

### Official Review · Reviewer_umCk · 2025-11-05

**Soundness:** 3
**Presentation:** 2
**Contribution:** 2
**Rating:** 4
**Confidence:** 4

**Summary:**

The paper develops a statistical learning framework for the proposed method, stating finite sample guarantees and excess risk bounds for the estimator. It highlights how regularity and beta-strong convexity assumptions drive stability and convergence rates.
The analysis connects optimization error and estimation error, with explicit dependence on sample size and ambient dimension. The results are positioned relative to statistical optimal transport, where dimension often controls rates.

**Strengths:**

Theoretical framing is clear, separating approximation, optimization, and estimation errors. The proofs use standard tools from empirical process theory and stability.

**Weaknesses:**

The beta strong convexity condition is assumed rather than derived from the model or data-generating process. Please either prove it under verifiable conditions or provide a constructive check that practitioners can apply to certify it.

**Questions:**

1. Under what concrete distributional conditions on the source and target does beta strong convexity hold?
2. The estimation error exhibits explicit dimensional dependence that resembles known lower bounds in statistical optimal transport. Can the authors clarify whether structures such as low intrinsic dimension, manifold support, or spectral decay can reduce the exponent, and if so, how this would change the bounds and constants?

---

> ### Author Response · Authors · 2025-11-22
> **Answer to Reviewer umCk (1/2)**
>
> Thank you for your thorough feedback. Please find the answers to your questions below.
>
> **[(W1) <...> The beta strong convexity condition is assumed rather than derived from the model or data-generating process. Please either prove it under verifiable conditions or provide a constructive check that practitioners can apply to certify it. <...>; (Q1) <...> Under what concrete distributional conditions on the source and target does beta strong convexity hold?]**
>
> Indeed, in our theoretical derivations we assume that the ground truth optimal potential $\varphi^*$ is strongly convex for some $\beta>0$. Importantly, this assumption is purely data-dependent; whether or not it holds depends solely on the source and target distributions, $p$ and $q$, and not on the particular way in which the OT problem is solved.
>
> Luckily, the regularity of the OT problem (especially with quadratic cost) is a well-studied topic, due to Caffarelli and other prominent researchers. In particular, following [1, Theorem ([C3])], it holds that
>
> - if the data supports $\mathcal{X}$, $\mathcal{Y}$ are convex;
>
> - probability distribution densities, $p$ and $q$ are strictly positive and bounded on $\mathcal{X}$ and $\mathcal{Y}$, respectively;
>
> - $p$ and $q$ are $C^{\alpha}$ for some $\alpha > 0$, i.e., Holder continuous functions: $\exists \alpha > 0$, $C > 0$, $\forall x_1, x_2 \in \mathcal{X}; \forall y_1, y_2 \in \mathcal{Y}$: $\vert p(x_1) - p(x_2) \vert \leq C \Vert x_1 - x_2 \Vert^{\alpha}$; $\vert q(y_1) - q(y_2) \vert \leq C \Vert y_1 - y_2 \Vert^{\alpha}$,
>
> then the optimal potential $\varphi^*$ is indeed strongly convex, and we can verifiably apply our developed theoretical framework.
>
> Noteworthy, the Holder continuity does not seem to be restrictive assumption: it holds for majority of "tabular" distributions, mixture of distributions (e.g., Gaussians), which are known to be universal approximators..
>
> Following the reviewer's suggestion, in the revised version of our manuscript, we added a Remark which concerns the assumption on the strong convexity of $\varphi^*$.
>
> [1] Caffarelli, L. A. (1996). Boundary regularity of maps with convex potentials--II. Annals of mathematics, 144(3), 453-496.
>
> **[(Q2) The estimation error exhibits explicit dimensional dependence that resembles known lower bounds in statistical optimal transport. Can the authors clarify whether structures such as low intrinsic dimension, manifold support, or spectral decay can reduce the exponent, and if so, how this would change the bounds and constants?]**
>
> We thank the reviewer for this interesting question. On the one hand, our rates indeed somehow depend on dimensionality of the data $d$. In particular, the rate $O(\frac{1}{\sqrt{N}})$ appearing in Corollary 4.9 (Corollary 4.10 in the revised version) hides the constant $\sqrt{d}$, where $d$ is the data dimension. On the other hand, compared to the statistical optimal transport bounds, e.g., those from (Hutter \& Rigollet, 2021), (Gunsilius, 2022), the dependence on the dimensionality does not affect the major sample rate ($\frac{1}{\sqrt{N}}$) w.r.t. the number of samples. While some recent studies in the direction of statistical optimal transport, e.g., (Divol et. al., 2025), (Ding et. al., 2024), indeed, as suggested by the reviewer, contain lower bounds which resemble ours, i.e., with major sample rate which is independent of dimension, and with (hidden) constant which may be affected (significantly or not significantly) by the dimension, this is done under a number of assumptions, and for the case of classical (semi-) dual OT, see eqs. (3), (4) of our revised manuscript. Our case much differs (and more difficult) - we analyze min-max OT, eq. (7), which prescribes optimizing a potential and a map (parameterized as neural networks) in an adversarial-like game. This limits the options to place assumptions to keep the result practical.
>
> In fact, the dependence of our derived estimation error on $d$ arises only in the constant; this dependence *directly* related to the usage of maps $T$ (neural networks) with $d$ - dimensional output, see eq. (62) in the revised manuscript. In principle, if we let neural networks operate in some ''latent'' space which defines a data manifold, we expect that the constant could be reduced accordingly. Still, this case (and similar ones, as noted by the reviewer: low intrinsic dimension, spectral decay) require a separate analysis and constitute an interesting direction for future research. Currently, the main message of our paper is that practically-inspiring min-max OT is *consistent* from statistical point of view and *follow* typical sample decay rates.

---

> ### Author Response · Authors · 2025-11-22
> **Answer to Reviewer umCk (2/2)**
>
> **References.**
>
> (Hutter \& Rigollet, 2021) Hütter, J. C., \& Rigollet, P. (2021). Minimax estimation of smooth optimal transport maps. The Annals of Statistics, 49(2), 1166.
>
> (Gunsilius, 2022) Gunsilius, F. F. (2022). On the convergence rate of potentials of Brenier maps. Econometric Theory, 38(2), 381-417.
>
> (Divol et. al., 2025) Divol, V., Niles-Weed, J., \& Pooladian, A. A. (2025). Optimal transport map estimation in general function spaces. The Annals of Statistics, 53(3), 963-988.
>
> (Ding et. al., 2024) Ding, Y., Li, R., \& Xue, L. (2024). Statistical convergence rates of optimal transport map estimation between general distributions. arXiv preprint arXiv:2412.08064.
>
> **Concluding remarks**
>
> Thank you for your thoughtful feedback. We would appreciate it if you could let us know whether our clarifications address your concerns. If there are any remaining issues, we would be happy to discuss them further. Otherwise, we kindly ask you to consider updating your score accordingly.

---

### Author Response · Authors · 2025-11-22
**General Response**

Dear Reviewers,

we are grateful for your constructive and insightful reviews!
We are encouraged by the fact that you acknowledge the well-written text of our paper (Reviewers BuZV, aRUC) and agree on the novelty and clarity of our theoretical results (Reviewers aRUC, 1LsA).

Below, we provide a list of the **main changes** incorporated into the *revised version of our paper*:

- **[Reviewers umCk, aRUC]** New remark 4.8 regarding the $\beta$-strong convexity assumption (*Section 4*);

- **[Reviewer BuZV]** Experiments with non-neural baseline estimators (*new Appendix C.1*);

- **[Reviewer BuZV]** Illustrations of the OT solver's results in experiments with approximation error (*new Appendix C.3*);

- **[Reviewer aRUC]** New high-dimensional experiments on the estimation error (*new Appendix C.2*);

- **[Reviewer 1LsA]** Shortening Section 2, removing some theoretical background on the OT with general cost function.

Apart from this, we fix the typos, include missing related work and incorporate necessary clarifications suggested by the reviewers. All changes in the revised version are highlighted in **blue color**.

Best regards,
the Authors

---

### Author Response · Authors · 2025-12-02
**Final comment**

Dear Area Chairs and Reviewers,

in this message, we would like to thank the Reviewers for their valuable suggestions. In our opinion, we have addressed all of the concerns raised in the reviews and regret that the Reviewers did not have time for continuing the discussion.

To aid the work of the Area Chairs, we would like to mention that the main changes in the revised version of our paper are listed in the 'General Response' below. In brief, most of the reviewers' concerns are related to the experiments supporting our theoretical results with only one question related to the theory, i.e., the restrictiveness of the $\beta$-strong convexity assumption. In our answers, we explain why this assumption holds true under some reasonable assumptions on the source and target distributions $p$ and $q$. Besides, while our paper focuses on theoretical analysis of the neural OT solver (from a statistical learning perspective), we have conducted all of the requested additional experiments as well.

Best regards,
the Authors

---

### Meta-Review · Area_Chair_svsf · 2026-01-04

**Summary:**

The reviewers recognize the theoretical contributions of the work on deriving generalization error bounds for quadratic min-max optimal transport (OT) solvers parameterized by neural networks.  Reviewers concerns on  beta strong convexity condition that is a key assumption is adequately addressed by the reviewers.  The concern related to the initial lack of  high-dimensional experiments on the estimation error is also address with new experiments by the authors. In addition the authors added  Illustrations of the OT solver's results in experiments with approximation error as well as new additional results with non-neural baseline estimators.  Overall, the authors have addressed most of the reviewer concerns.  My assessment for this paper is that the derivation  and analysis on the generaization bounds and their decomposition into estimation and approximation errors has scientific merit.  The paper is overall well written and it would be a useful contribution to the ML community.

**Reviewer Concerns:**

Most concerns have been addressed by the authors. I think one interesting debate is the question Q4 on the last reviewer 1LsA. Overall I find the answer of the authors convincing.

**Reviewer Scores:**

Reviewer umCk would have increase its score to 6 and above. The concern on beta strong convexity condition together with the questions  are addressed by the authors.

Reviewer  BuZV would have maintain its score to 6  or increase it.

Reviewer aRUC would have maintain its score to 6  or increase it.

Reviewer 1LsA admits that most concerns are addressed. I can not see a reason for why  not increasing the score.

---

### Decision · Program_Chairs · 2026-01-26

Accept (Poster)